# NoisyRollout: Reinforcing Visual Reasoning with Data Augmentation

Xiangyan Liu[*1]    Jinjie Ni[*1]    Zijian Wu[*1]    Chao Du[2]    Longxu Dou[2]    Haonan Wang[1]
Tianyu Pang[†2]    Michael Qizhe Shieh[1]

[1]National University of Singapore    [2]Sea AI Lab    ⚙ **Code**    🤗 **Collection**

## Abstract

Recent advances in reinforcement learning (RL) have strengthened the reasoning capabilities of vision-language models (VLMs). However, enhancing policy exploration to better scale test-time compute remains largely underexplored. In addition, VLMs continue to struggle with imperfect visual perception, which in turn affects the subsequent reasoning process. We introduce **NoisyRollout**, a simple yet effective data augmentation method that addresses these issues by mixing training trajectories from both clean and moderately distorted images. This approach injects perceptual diversity, encouraging better policy exploration and leading to more robust reasoning. A noise annealing schedule gradually reduces distortion strength, aiding exploration early in training while ensuring later stability. Crucially, our method is easy-to-adopt—**requiring no additional training cost and no modifications to the RL objective**. Extensive experiments on **2** distinct training datasets demonstrate that NoisyRollout achieves state-of-the-art performance among open-source RL-tuned models across **5** out-of-domain reasoning and perception benchmarks. Furthermore, we validate the effectiveness of NoisyRollout across model sizes (7B and 32B), data scales (from 1K to 6K) and image augmentation types (Gaussion noise and rotation), highlighting its generalizability and scalability.

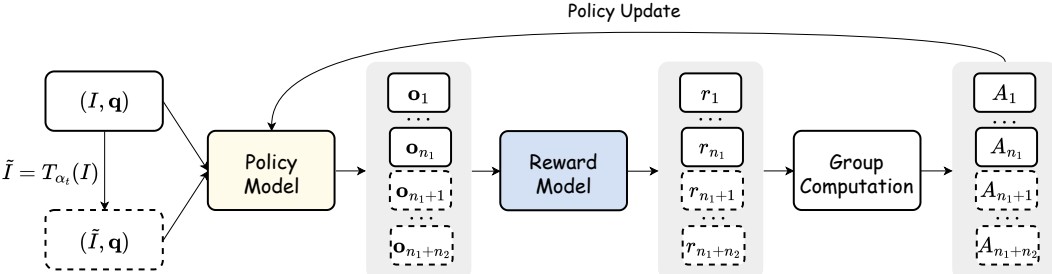

Figure 1: An Illustration of the NoisyRollout workflow. Solid lines depict the generation and use of clean rollouts from the clean (original) input $(I, \mathbf{q})$, while dashed lines depict the generation and use of noisy rollouts from the corresponding noisy input $(\tilde{I}, \mathbf{q})$. The distorted image $\tilde{I}$ is obtained by applying a distortion function $\tilde{I} = T_{\alpha_t}(I)$ with distortion strength $\alpha_t$. The distortion level $\alpha_t$ is controlled by a noise annealing schedule, which gradually decreases distortion during training. Rollouts from both sources are mixed to form the final trajectories $\{\mathbf{o}_i\}_{i=1}^{n_1+n_2}$, rewards $\{r_i\}_{i=1}^{n_1+n_2}$, and advantages $\{A_i\}_{i=1}^{n_1+n_2}$. Crucially, policy optimization conditions only on clean inputs $(I, \mathbf{q})$; the corresponding noisy inputs $(\tilde{I}, \mathbf{q})$ are used solely to collect diverse rollouts for exploration.

---

[*]Equal contribution. [†]Correspondence to Tianyu Pang.

39th Conference on Neural Information Processing Systems (NeurIPS 2025).

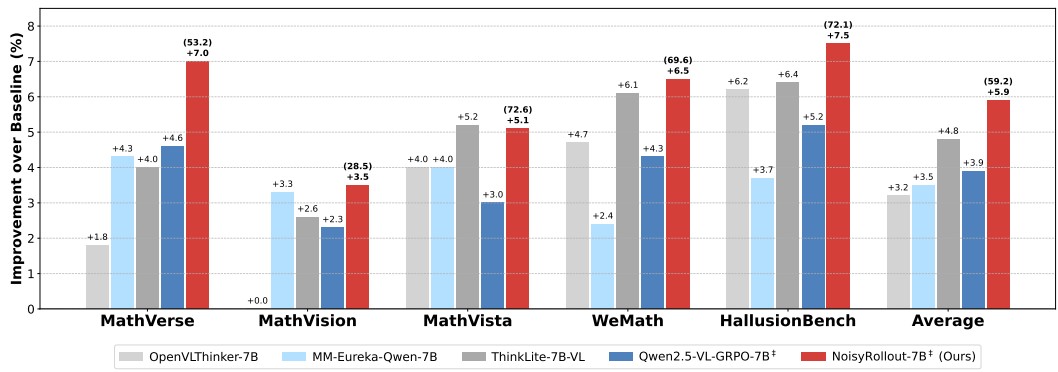

Figure 2: Accuracy improvement over Qwen2.5-VL-7B-Instruct on 5 out-of-domain benchmarks, covering both visual reasoning tasks (from MathVerse to WeMath) and a visual perception task (HallusionBench). Both Qwen2.5-VL-GRPO-7B and NoisyRollout-7B are fine-tuned by ourselves (denoted with ‡) using vanilla GRPO with only **2.1K** training samples from Geometry3K. The exact accuracy of NoisyRollout-7B is annotated above each corresponding bar in parentheses.

# 1 Introduction

Scaling test-time compute—often referred to as *reasoning*—through reinforcement learning (RL) has emerged as a promising axis for advancing model intelligence [25, 11]. While this idea has been primarily explored in the context of large language models (LLMs) [21, 90], the vision-language model (VLM) community is also actively investigating this direction [46, 56, 53]. Recent endeavours suggests that VLMs can also benefit from RL-driven scaling of test-time compute [23, 44, 72, 49, 86].

However, scaling test-time compute via RL requires more than sheerly generating longer outputs [45], and VLMs face unique challenges in this process. A key challenge is effective policy exploration, enabling policies to discover behaviors that generalize well beyond training data [87, 82]—an area largely underexplored in VLM research. Traditional practices, such as increasing rollout temperature to promote decoding diversity [89], often introduce superficial variability without meaningfully directing policies toward more robust or informative behaviors.

Moreover, VLMs inherently struggle with imperfect visual perception [40, 78], which negatively impacts subsequent reasoning processes [93, 97, 27]. Despite this, recent efforts [46, 53, 13] tend to adapt RL methods directly from the LLM domain. Such approaches often fail to take these perceptual challenges into consideration, thereby hindering the efficient development of visual reasoning capabilities through RL training.

Tackling the challenges of policy exploration and perceptual limitations in VLMs during RL training, we propose **NoisyRollout**, a simple yet powerful data augmentation technique for VLMs that introduces *meaningful rollout diversity*. Specifically, for each training sample consisting of an input image $I$ and a corresponding text query $\mathbf{q}$, the old policy ($\pi_{\theta_{old}}$) produces two sets of rollouts based on the original clean image and a moderately distorted version of the same image, respectively.

While the current policy ($\pi_{\theta}$) is updated solely by conditioning on the clean image and text query pair $(I, \mathbf{q})$, the two sets of rollouts form a group, collectively contributing to computing the reward baseline and normalized advantage in Group Relative Policy Optimization (GRPO) [64]. This *hybrid rollout strategy* enables the policy to achieve more targeted and efficient exploration, ultimately leading to *more robust visual reasoning* via RL through two key mechanisms:

❶ Successful reasoning trajectories from noisy inputs with distorted images reveal alternative, potentially more robust reasoning strategies, **improving reasoning generalization to harder or out-of-domain visual conditions**.

❷ When the same query yields different outcomes for clean and distorted inputs, the resulting reward differences expose *perceptual discrepancies* that affect reasoning. These discrepancies act as implicit contrastive signals, **helping refine the model's visual perception during reasoning** by constraining the negative perceptual exploration space.

While incorporating noisy rollouts can facilitate more effective and efficient exploration, it may also introduce instability in policy gradient estimation. To further enhance scalability and training stability, we employ a *noise annealing schedule* that gradually reduces the strength of image distortions over training. Such a strategy mitigates distributional mismatch between the evolving policy and the noisy trajectories generated from it when conditioned on clean inputs—an issue that often arises in later training stages—while retaining the benefits of noisy signals during the early phases of training.

We conduct extensive experiments to validate the effectiveness of NoisyRollout. Trained with only 2.1K samples from the Geometry3K [47] dataset using Qwen2.5-7B-VL-Instruct [4], Figure 2 shows that NoisyRollout achieves superior performance across 5 out-of-domain visual reasoning and perception benchmarks [48, 19, 93, 74, 57] (MathVerse 53.2%, MathVision 28.5%, and HallusionBench 72.1%). It outperforms both open-source RL-tuned models [53, 77] and those utilizing large-scale supervised fine-tuning (SFT) before RL [83, 92, 13]. Furthermore, it consistently surpasses its direct baseline (vanilla GRPO) on both in-domain and out-of-domain tasks, all within a fixed total rollout budget. Crucially, these out-of-domain improvements generalize across different model sizes (e.g., 7B to 32B) as well as training corpora and data scales (e.g., MMK12 [53] with 1K to 6K samples). These empirical results, combined with its simplicity and lightweight characteristics, establish NoisyRollout as a potentially *scalable* approach.

## 2 NoisyRollout: A Free-Lunch with Noisy Reinforcement Learning

We introduce NoisyRollout, a data augmentation method that enhances visual reasoning in VLMs during RL training, particularly by improving the rollout diversity for better policy exploration. NoisyRollout achieves this by incorporating a hybrid rollout strategy that leverages reasoning trajectories from both clean and distorted images, and a noise annealing schedule that progressively reduces distortion strength. These designs require no additional training cost and integrate seamlessly with standard GRPO implementations. A simplified overview is provided in Figure 1 and Algorithm 1.

**GRPO.** Group Relative Policy Optimization (GRPO) [64] was originally developed to improve mathematical reasoning in LLMs but can also be effectively adapted to enhance visual reasoning in VLMs. For a given input pair $(I, \mathbf{q})$ consisting of an image and text query from the training set $p_{\mathcal{D}}$, a rule-based outcome reward function $r(I, \mathbf{q}, \mathbf{o})$ is adopted to avoid reward hacking. This function assigns $r(I, \mathbf{q}, \mathbf{o}) = 1$ if the generated response $\mathbf{o}$ correctly addresses the query (as verified by a parser) with the required format, and $r(I, \mathbf{q}, \mathbf{o}) = 0$ otherwise. For each input, the old policy $\pi_{\theta_{\text{old}}}$ generates $n$ response rollouts. The baseline reward is then calculated as $\text{mean}(\mathbf{r})$, where $\mathbf{r} = \{r_i\}_{i=1}^{n} = \{r(I, \mathbf{q}, \mathbf{o}_i)\}_{i=1}^{n}$ represents the rewards for all rollouts. The normalized advantage for the $i$-th rollout is defined as $\hat{A}_i = \frac{r_i - \text{mean}(\mathbf{r})}{\text{std}(\mathbf{r})}$. Derived from PPO [62], the GRPO objective function is:

$$\mathcal{J}_{\text{GRPO}}(\theta) = \mathbb{E}_{(I,\mathbf{q}) \sim p_{\mathcal{D}}, \mathbf{o} \sim \pi_{\theta_{\text{old}}}(\cdot | I, \mathbf{q})}$$
$$\left[ \frac{1}{n} \sum_{i=1}^{n} \min \left( \frac{\pi_\theta(\mathbf{o}_i \mid I, \mathbf{q})}{\pi_{\theta_{\text{old}}}(\mathbf{o}_i \mid I, \mathbf{q})} \hat{A}_i, \text{clip}\left( \frac{\pi_\theta(\mathbf{o}_i \mid I, \mathbf{q})}{\pi_{\theta_{\text{old}}}(\mathbf{o}_i \mid I, \mathbf{q})}, 1 - \epsilon, 1 + \epsilon \right) \hat{A}_i \right) \right], \quad (1)$$

where $\pi_\theta$ is the current policy, $\epsilon > 0$ sets the clipping range. We omit the KL divergence constraint $\mathbb{D}_{\text{KL}}[\pi_\theta | \pi_{\theta_{\text{ref}}}]$ following recent practices in Meng et al. [53] and Liu et al. [45].

**Hybrid rollout strategy.** Building upon GRPO, NoisyRollout introduces a hybrid rollout strategy to enhance the rollout diversity. For each input pair $(I, \mathbf{q})$, we generate an augmented version of the image $\tilde{I}$ through a noise transformation function $T_\alpha$ parameterized by a distortion strength $\alpha$, i.e., $\tilde{I} = T_\alpha(I)$. As illustrated in Figure 1, the old policy $\pi_{\theta_{\text{old}}}$ produces two sets of rollouts: $n_1$ responses conditioned on the clean input $(I, \mathbf{q})$, and $n_2$ responses conditioned on the corresponding noisy input $(\tilde{I}, \mathbf{q})$. All rollouts from both clean and distorted images are then combined into a single group for reward calculation, yielding $\mathbf{r} = \{r_i\}_{i=1}^{n_1+n_2} = \{r(I, \mathbf{q}, \mathbf{o}_j)\}_{j=1}^{n_1} \cup \{r(I, \mathbf{q}, \mathbf{o}_k)\}_{k=n_1+1}^{n_1+n_2}$. Crucially, the policy update step remains conditioned solely on the clean image $I$ and query $\mathbf{q}$ for better policy exploration. We defer the discussion of optimizing noisy and clean trajectories on their

---

**Algorithm 1** NoisyRollout: Noisy Reinforcement Fine-Tuning

---

1: **Input:** Current policy $\pi_\theta$, old policy $\pi_{\theta_{old}}$, dataset $p_\mathcal{D}$, training steps $t_{\max}$, clean rollout number $n_1$, noisy rollout number $n_2$, clip parameter $\epsilon$, initial noise strength $\alpha_0$, noise scheduler $\eta(\cdot)$, noise transformation function $T(\cdot)$

2: **for** $t = 1$ to $t_{\max}$ **do**

3:      Sample batch $(I, \mathbf{q}) \sim p_\mathcal{D}$

4:      Set noise strength $\alpha_t = \eta(\alpha_0, t, t_{\max})$                  ▷ Annealing schedule

5:      Generate distorted images $\tilde{I} = T_{\alpha_t}(I)$

6:      Sample $\{\mathbf{o}_j\}_{j=1}^{n_1}$ from $\pi_{\theta_{old}}(\mathbf{o} \mid I, \mathbf{q})$             ▷ Clean rollouts

7:      Sample $\{\mathbf{o}_k\}_{k=n_1+1}^{n_1+n_2}$ from $\pi_{\theta_{old}}(\mathbf{o} \mid \tilde{I}, \mathbf{q})$       ▷ Noisy rollouts

8:      Compute rewards $r_i = r(I, \mathbf{q}, \mathbf{o}_i)$ for all $i \in \{1, \ldots, n_1 + n_2\}$

9:      Compute advantages $\hat{A}_i = \frac{r_i - \text{mean}(\mathbf{r})}{\text{std}(\mathbf{r})}$, where $\mathbf{r} = \{r_i\}_{i=1}^{n_1+n_2}$

10:     Update policy using:

11:     $\mathcal{J}(\theta) = \mathbb{E}\left[\frac{1}{n_1+n_2}\sum_{i=1}^{n_1+n_2} \min\left(\frac{\pi_\theta(\mathbf{o}_i|I,\mathbf{q})}{\pi_{\theta_{old}}(\mathbf{o}_i|I,\mathbf{q})}\hat{A}_i, \text{clip}\left(\frac{\pi_\theta(\mathbf{o}_i|I,\mathbf{q})}{\pi_{\theta_{old}}(\mathbf{o}_i|I,\mathbf{q})}, 1-\epsilon, 1+\epsilon\right)\hat{A}_i\right)\right]$

12:     $\theta \leftarrow \theta - \nabla_\theta \mathcal{J}(\theta)$           ▷ Update conditioned on clean images only

13:     $\theta_{old} \leftarrow \theta$                      ▷ Update old policy parameters

14: **end for**

---

corresponding inputs to Appendix B. The NoisyRollout objective function is defined as:

$$\mathcal{J}(\theta) = \mathbb{E}_{(I,\mathbf{q})\sim p_\mathcal{D}, \{\mathbf{o}_j\}_{j=1}^{n_1}\sim\pi_{\theta_{old}}(\cdot|I,\mathbf{q}), \{\mathbf{o}_k\}_{k=n_1+1}^{n_1+n_2}\sim\pi_{\theta_{old}}(\cdot|\tilde{I},\mathbf{q})}$$

$$\left[\frac{1}{n_1+n_2}\sum_{i=1}^{n_1+n_2}\min\left(\frac{\pi_\theta(\mathbf{o}_i \mid I,\mathbf{q})}{\pi_{\theta_{old}}(\mathbf{o}_i \mid I,\mathbf{q})}\hat{A}_i, \text{clip}\left(\frac{\pi_\theta(\mathbf{o}_i \mid I,\mathbf{q})}{\pi_{\theta_{old}}(\mathbf{o}_i \mid I,\mathbf{q})}, 1-\epsilon, 1+\epsilon\right)\hat{A}_i\right)\right]. \quad (2)$$

**Noise annealing schedule.** Applying fixed-strength distortions throughout training often leads to training instability, primarily due to a distributional mismatch between noisy rollouts and the evolving policy. To mitigate this, we introduce a noise annealing schedule $\eta(\cdot)$ that gradually reduces the distortion strength over time. Specifically, at training step $t$, the noise level is defined as $\alpha_t = \eta(\alpha_0, t, t_{\max})$, where $\alpha_0$ is the initial noise strength and $t_{\max}$ denotes the total number of training steps. As shown in Figure 1, the distorted image is then generated as $\tilde{I} = T_{\alpha_t}(I)$.

Consequently, this schedule keeps diverse and informative supervision signals early in training, when the policy is constrained by its perceptual capacity. As training progresses, the noise level $\alpha_t$ is gradually reduced, narrowing the gap between noisy rollouts ($\{\mathbf{o}_k\}_{k=n_1+1}^{n_1+n_2}$) and the trajectories that $\pi_{\theta_{old}}(\cdot|I,\mathbf{q})$ would typically produce. This decay helps mitigate abrupt distribution shifts after policy updates, which can arise from unstable or high-variance policy gradients. Over time, rollouts generated from $(\tilde{I}, \mathbf{q})$ become progressively more "on-policy" *w.r.t* the clean-input-conditioned policy $\pi_{\theta_{old}}(\cdot|I,\mathbf{q})$, fostering a smoother transition from exploration to exploitation in later training stages.

**Summary.** NoisyRollout aims to improve the visual reasoning abilities of VLMs by enhancing rollout diversity to enable more effective policy exploration during RL training. Built on top of GRPO, it introduces a *hybrid rollout strategy* and a *noise annealing schedule*. These additions require no extra training cost and preserve the original RL objective. This design offers several benefits:

- **Robust reasoning:** Positive trajectories[1] from distorted inputs offer alternative, and potentially more robust reasoning paths, improving generalization to challenging or out-of-domain visual conditions.
- **Contrastive perceptual signals:** When clean and distorted inputs yield divergent outcomes for the same text query, the resulting reward differences shape a better perceptual exploration space, serving as implicit contrastive signals that refine the model's perceptual behaviors during reasoning.
- **Stable training dynamics for better exploitation:** The noise annealing schedule enables a smooth transition from early-stage noisy signals to fully on-policy learning, mitigating distributional mismatch and ensuring stable convergence as the model gradually improves its perception and reasoning. This provides a solid foundation for further exploitation in the later stages of RL training.

---

[1]We regard a trajectory as positive if it receives a reward of 1.

Table 1: Performance comparison of VLMs with moderate parameter sizes on a suite of out-of-domain benchmarks. Accuracy scores (%) are reported for all benchmarks for clarity. Models marked with "*" are evaluated using our evaluation suite. For R1-related models, the corresponding `reasoning` templates are used by default, while "†" indicates results obtained using the `direct-answer` template. Data sizes used for SFT and RL are annotated in blue and red, respectively. The best value in each column is shown in **bold**, and the second-best is underlined.

| Model | Data Size | MathVerse | MathVision | MathVista | WeMath | HallusionBench |
|---|---|---|---|---|---|---|
| *Open-source* | | | | | | |
| InternVL-2.5-8B-Instruct [9] | - | 39.5 | 19.7 | 64.4 | - | 67.3† |
| LLaVA-OneVision-7B [35] | - | 26.2 | - | 63.2 | - | 48.4† |
| Kimi-VL-16B [29] | - | 44.9 | 21.4 | 68.7 | - | 66.2† |
| URSA-8B [50] | - | 45.7 | 26.2 | 59.8 | - | - |
| Mulberry-7B [84] | - | - | - | 63.1 | - | - |
| *R1-related (reinforcement learning with verifiable reward)* | | | | | | |
| R1-VL-7B [92] | 260K+10K | 40.0 | 24.7 | 63.5 | - | - |
| Vision-R1-7B [23] | 200K+10K | 52.4 | - | 73.5 | - | - |
| R1-OneVision-7B* [83] | 155K+10K | 46.1 | 22.5 | 63.9 | 62.1 | 65.6 |
| OpenVLThinker-7B* [13] | 35K+15K | 48.0 | 25.0 | 71.5 | 67.8 | 70.8 |
| MM-Eureka-Qwen-7B* [53] | 15K | 50.5 | 28.3 | 71.5 | 65.5 | 68.3 |
| ADORA-7B* [20] | 2.1K | 50.1 | 27.6 | 71.1 | 67.1 | 53.1 |
| ThinkLite-7B-VL* [77] | 11K | 50.2 | 27.6 | 72.7 | 69.2 | 71.0 |
| VLAA-Thinker-7B* [5] | 25K | 49.9 | 26.9 | 68.8 | 67.9 | 68.6 |
| Qwen2.5-VL-7B-Instruct* [4] | - | 46.2 | 25.0 | 67.5 | 63.1 | 64.6 (71.2†) |
| + Vanilla GRPO* ($n = 12$) | 2.1K (Geometry3K) | 50.8 | 27.3 | 70.5 | 67.4 | 69.8 |
| + NoisyRollout* ($n_1 = 6, n_2 = 6$) | 2.1K (Geometry3K) | **53.2** | 28.5 | 72.6 | 69.6 | 72.1 |
| + Vanilla GRPO* ($n = 12$) | 6.4K (MMK12) | 51.8 | 29.4 | 73.2 | 70.2 | 70.3 |
| + NoisyRollout* ($n_1 = 6, n_2 = 6$) | 6.4K (MMK12) | 53.0 | **30.6** | **74.5** | **70.3** | **72.2** |

Table 2: Performance comparison of VLMs with large parameter sizes on a suite of out-of-domain benchmarks. The notation and evaluation protocols are consistent with those described in Table 1.

| Model | #Data | MathVerse | MathVision | MathVista | WeMath | HallusionBench |
|---|---|---|---|---|---|---|
| *Close-source* | | | | | | |
| GPT-4o [24] | - | 50.8 | 30.4 | 63.8 | 69.0 | 71.4† |
| Claude-3.5-Sonnet [3] | - | 26.5 | 38.0 | 67.7 | - | 71.6† |
| Kimi1.5 [28] | - | - | 38.6 | 74.9 | - | - |
| *Open-source* | | | | | | |
| InternVL-2.5-78B-Instruct [9] | - | 51.7 | 32.2 | 72.3 | - | 72.9† |
| QVQ-72B-Preview [58] | - | - | 35.9 | 71.4 | - | - |
| Qwen2.5-VL-72B-Instruct [4] | - | - | 38.1 | 74.8 | - | 71.9† |
| *R1-related (RL-tuned with verifiable reward)* | | | | | | |
| MM-Eureka-Zero-38B [53] | 9.4K | 48.9 | 26.6 | 64.2 | - | - |
| MM-Eureka-Qwen-32B* [53] | 17K | 56.5 | 39.8 | 76.7 | 76.7 | 71.4 |
| Qwen2.5-VL-32B-Instruct* [4] | - | 58.5 | 37.6 | 76.5 | 74.0 | 66.6 |
| + Vanilla GRPO* ($n = 8$) | 2.1K (Geometry3K) | 58.9 | 39.2 | 77.0 | 76.1 | 72.3 |
| + NoisyRollout* ($n_1 = 4, n_2 = 4$) | 2.1K (Geometry3K) | 58.9 | 39.9 | **77.8** | 77.2 | **73.5** |
| + Vanilla GRPO* ($n = 8$) | 6.4K (MMK12) | 58.9 | 40.0 | 76.7 | 76.9 | 72.1 |
| + NoisyRollout* ($n_1 = 4, n_2 = 4$) | 6.4K (MMK12) | **59.3** | **41.6** | 77.4 | **77.6** | 73.2 |

## 3 Experiments

**Dataset.** We use EasyR1 [96] as our reinforcement learning training framework, which is built on verl [65] and specifically designed for VLMs. Our experiments utilize two datasets: Geometry3K [47], focused on geometric problem solving, and MMK12 [53], covering diverse K-12 math topics. These datasets comprise 2.1K and 6.4K training samples respectively. We processed them by converting all questions from multiple-choice to free-form format to prevent reward hacking and model guessing.

**Evaluation.** We mainly evaluate model performance along two dimensions. First, we assess out-of-domain generalization across five benchmarks: four visual reasoning benchmarks, including MathVerse [93], MathVision [74], MathVista [48], and WeMath [57], as well as one visual perception

benchmark, HallusionBench [19]. Second, we evaluate the in-domain performance of NoisyRollout by comparing it with the vanilla GRPO baseline on the Geometry3K test set.

Moreover, we develop an evaluation suite for consistent assessment of our trained checkpoints and most open-source R1-related checkpoints using vLLM [31] for accelerated inference (marked with $\star$ in Tables 1 and 2), while adopting reported results for others.[2] We employ *greedy decoding* for model inference and use Gemini-2.0-Flash-001 [17] as the judge model to parse generated responses.

**Implementation details.** Following prior work [53, 77], we initialize our policy models with Qwen2.5-VL-7/32B-Instruct, which exhibit strong foundational capabilities well-suited for subsequent RL training. All experiments are conducted using 8 A100 GPUs (40G for 7B model, 80G for 32B model). We keep the vision encoder frozen for training stability and parameter efficiency. For other general RL-related hyperparameters, we adopt the default settings from EasyR1: a global batch size of 128, a rollout batch size of 512, a rollout temperature of 1.0, and a learning rate of $1e{-}6$. To prevent token-length bias, we compute the policy loss using the `token-mean` aggregation strategy.[3] For NoisyRollout-specific configurations, we adopt **Gaussian noise** as the default image distortion strategy, and apply a *sigmoid-shaped* annealing schedule:

$$\alpha_t = \eta(\alpha_0, t, t_{\max}) = \alpha_0 \cdot \left( 1 - \frac{1}{1 + e^{-\lambda(t-\gamma)/t_{\max}}} \right), \tag{3}$$

where $\gamma$ determines the midpoint of the annealing curve and $\lambda$ controls its steepness. Figure 7 illustrates the visual effects of applying different levels of Gaussian noise to a clean image. We defer the discussion of unsuccessful image distortion strategies (e.g., cropping), noise annealing strategies (e.g., power, exponential), and proportions of noisy rollouts in total rollouts to Appendix A. The `reasoning` and `direct-answer` templates used in our experiments are shown in Appendix H. Additional implementation details regarding the number of training steps/epochs and the hyperparameters for image distortion and noise annealing are presented in Appendix J.

### 3.1 Main Results

**Result 1: Out-of-domain generalization.** When trained on the Geometry3K dataset using Qwen2.5-VL-7B-Instruct, NoisyRollout not only improves in-domain performance (Figure 3, lower left subplot), but more importantly, demonstrates strong out-of-domain generalization. As shown in Table 1, NoisyRollout achieves superior performance across five visual reasoning and perception benchmarks, consistently outperforming the vanilla GRPO baseline in every case. This advantage is further illustrated in Figure 3, which presents detailed comparisons across benchmarks as training progresses. Specifically, NoisyRollout achieves 53.2% on MathVerse, 28.5% on MathVision, and 69.6% on WeMath, surpassing existing R1-related baselines and even outperforming GPT-4o.

Moreover, while Qwen2.5-7B-VL-Instruct's perception accuracy on HallusionBench drops from 71.2% to 64.6% when switching from `direct-answer` to `reasoning` templates,[4] NoisyRollout achieves 72.1% with the `reasoning` prompt (compared to vanilla GRPO's 69.8%). The final subplot in Figure 3 further confirms that NoisyRollout enhances perception quality during reasoning, achieving a higher Bradley–Terry win rate over vanilla GRPO (See Appendix C for details). These results indicate that our hybrid rollout strategy enhances visual perception by promoting better policy exploration through vision-oriented inductive biases.

**Result 2: Sample efficiency.** NoisyRollout demonstrates exceptional data efficiency by generalizing with only **2.1K** training samples from Geometry3K, whereas comparable models require significantly more data or even additional SFT as warm-up training. For example, Table 1 indicates that OpenVLThinker-7B needs **35K** SFT samples and **15K** RL samples but reaches only 48.0% on MathVerse and 71.5% on MathVista. This efficiency stems from NoisyRollout's use of noisy training signals that foster targeted exploration during RL, enabling effective generalization from limited samples.

**Result 3: Robustness across training datasets and model sizes.** NoisyRollout consistently improves upon vanilla GRPO, demonstrating strong robustness across model sizes and training datasets. As shown in Tables 1 and 2, the 7B model trained on MMK12 achieves gains of 1.2%, 1.3%, and

---

[2]While we closely follow system (or format) prompts from relevant codebases or papers, minor result discrepancies may occur due to differences in judge models or inference engines, which we consider acceptable.

[3]The code implementation can be found at verl.

[4]This degradation caused by the `reasoning` template has also been observed in previous studies [6, 27].

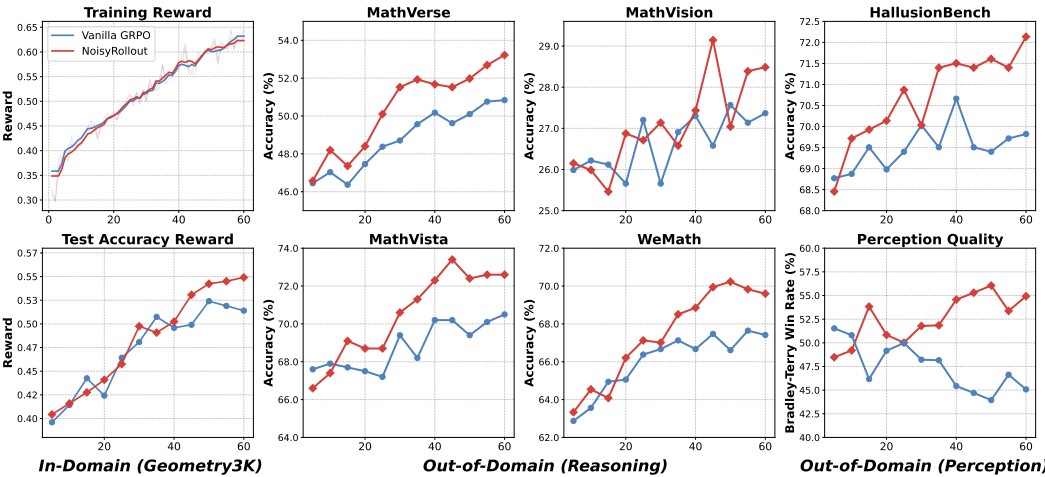

Figure 3: Comparison of NoisyRollout and vanilla GRPO on Qwen2.5-VL-7B-Instruct across in-domain and out-of-domain scenarios with the same total rollout number (12). The X-axis in all subplots represents *RL training steps*. **First column:** Reward comparison on the in-domain dataset during training. **Second and third columns:** Comparison on four out-of-domain visual reasoning benchmarks. **Last column:** Evaluation of visual perception capabilities, where the upper subplot directly compares their perception performance on HallusionBench and the lower subplot presents the model-ranked Bradley–Terry win rates *w.r.t.* the perception qualities of their reasoning traces.

1.9% over GRPO on MathVerse, MathVista, and HallusionBench, respectively. Similarly, the 32B model trained on MMK12 surpasses GRPO by 0.4%, 0.7%, and 1.1% on the same benchmarks.

Notably, in certain benchmarks like MathVerse and MathVista, the performance gains of NoisyRollout over vanilla GRPO are smaller for the 32B model than for the 7B model. This is likely because the 32B model's initial policy was already fine-tuned via RL,[5] whereas the 7B model's was not.

## 3.2 Ablation Study: More Effective Rollout Diversity with Noisy Trajectories

**Setup.** Unless otherwise specified, all ablation studies in this and the following subsection use Geometry3K as the training dataset on Qwen2.5-VL-7B-Instruct. In this part, we aim to examine the effectiveness of our NoisyRollout from the perspective of *rollout diversity*, a key factor for effective policy exploration in RL training. Here, we define rollout diversity as the average pairwise cosine distance between trajectory embeddings, where higher values indicate greater diversity. We randomly sample 256 instances from the Geometry3K training set. For each sample, we generate either $n = 12$ trajectories in vanilla GRPO or a combination of $n_1 = 6$ and $n_2 = 6$ trajectories in NoisyRollout, then encode them with an embedding model.[6] We track both diversity and accuracy across training steps (Figure 4) and evaluate final performance on in-domain and out-of-domain benchmarks (Table 3). We use vanilla GRPO with a rollout temperature of 1.0 as *the control group*.

**Result.** As shown in Figure 4, NoisyRollout enhances rollout diversity in early training stages compared to the control group, similar to increasing rollout temperature in vanilla GRPO from 1.0 to 1.2. This initial diversity boost, though accompanied by lower starting accuracy, ultimately leads to higher final training accuracy. Moreover, both NoisyRollout and higher-temperature vanilla GRPO show diversity decreasing below the control group in later training stages.

Table 3 reveals that NoisyRollout with temperature 1.0 consistently outperforms vanilla GRPO across all temperature settings (0.8 to 1.4), as well as mixed-temperature variants. Moreover, when applying temperature 1.2 to both approaches, NoisyRollout still demonstrates significant improvement over vanilla GRPO. These results indicate that NoisyRollout introduces more targeted and effective diversity than simply adjusting temperature parameters, which increases diversity in a less focused manner.

---

[5] https://qwenlm.github.io/blog/qwen2.5-vl-32b/
[6] https://huggingface.co/sentence-transformers/all-MiniLM-L6-v2

Table 3: Performance comparison under different rollout temperature settings, with the total number of rollouts fixed at 12. **In vanilla GRPO**, "$n(6):1.0$, $n(6):1.2$" indicates 6 rollouts with temperature 1.0 and another 6 with temperature 1.2. **In NoisyRollout**, "$n_1(6):1.0$" denotes 6 rollouts per sample generated from clean input $(I, \mathbf{q})$ with temperature 1.0, while "$n_2(6):1.0$" denotes 6 rollouts per sample from noisy input $(\tilde{I}, \mathbf{q})$ with temperature 1.0. "Geo3K" represents the test set of Geometry3K dataset. "Avg." represents average accuracy (%) across six benchmarks.

| Method | Rollout Temperature | Geo3K | MathVerse | MathVision | MathVista | WeMath | HallusionBench | Avg. |
|---|---|---|---|---|---|---|---|---|
| Vanilla GRPO | $n(12):0.8$ | 50.1 | 50.5 | 26.7 | 69.9 | 65.8 | 70.1 | 55.5 |
| | $n(12):1.0$ | 51.4 | 50.8 | 27.3 | 70.5 | 67.4 | 69.8 | 56.2 |
| | $n(12):1.1$ | 50.4 | 50.2 | 27.7 | 70.4 | 68.1 | 69.4 | 56.0 |
| | $n(12):1.2$ | 53.2 | 51.2 | 27.1 | 69.3 | 68.3 | 70.9 | 56.7 |
| | $n(12):1.4$ | 51.4 | 50.6 | 25.8 | 70.1 | 69.0 | 69.6 | 56.1 |
| | $n(6):1.0, n(6):1.2$ | 50.8 | 50.7 | 26.8 | 70.1 | 67.4 | 68.2 | 55.7 |
| NoisyRollout | $n_1(6):1.0, n_2(6):1.0$ | **54.9** | **53.2** | **28.5** | 72.6 | 69.6 | **72.1** | **58.5** |
| | $n_1(6):1.2, n_2(6):1.2$ | 53.4 | 52.6 | 28.3 | **72.9** | **70.9** | 70.9 | 58.2 |

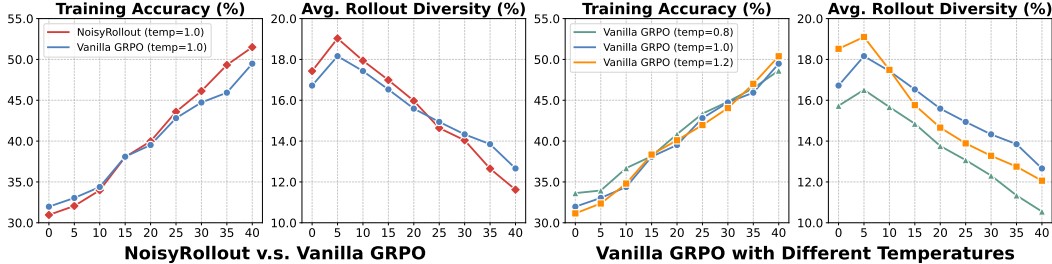

Figure 4: Comparison of accuracy and diversity metrics (%) across RL training steps (0 to 40). The left two subfigures contrast NoisyRollout versus vanilla GRPO (both with temperature 1.0), while the right two demonstrate the effects of different temperature settings (0.8, 1.0, 1.2) on vanilla GRPO.

## 3.3 Ablation Study: Impact of Hyperparameters and Module Design

**Noise annealing.** As shown in Figure 5, removing noise annealing causes the in-domain performance of our method to drop sharply around training step 45. This drop is due to divergence caused by a distributional mismatch—an issue discussed in Section 2 and further illustrated by the training dynamics in the same figure. Additionally, Table 4 shows that disabling noise annealing leads to lower performance in both in-domain and out-of-domain settings (43.9% and 58.0%, respectively), compared to our standard setting with noise annealing (54.9% and 59.2%). These results further highlight the effectiveness of noise annealing.

Table 4: Ablation study on the noise annealing strategy.

| Method | Geometry3K | OOD Avg. |
|---|---|---|
| Qwen2.5-VL-7B-Instruct | 39.4 | 53.3 |
| + Vanilla GRPO | 51.4 | 57.2 |
| + NoisyRollout w.o. Noise Annealing | 43.9 | 58.0 |
| + NoisyRollout | **54.9** | **59.2** |

Table 5: Empirical validation with additional augmentation types.

| Method | Augmentation | MathVerse | MathVision | MathVista | WeMath | HallusionBench | Average |
|---|---|---|---|---|---|---|---|
| GRPO | None | 50.8 | 27.3 | 70.5 | 67.4 | 69.8 | 57.2 |
| NoisyRollout | Gaussian noise | **53.2** (+2.4) | **28.5** (+1.2) | **72.6** (+2.1) | **69.6** (+2.2) | **72.1** (+2.3) | **59.2** (+2.0) |
| NoisyRollout | Rotation | 52.5 (+1.7) | 28.1 (+0.8) | 71.9 (+1.4) | 68.1 (+0.7) | 70.2 (+0.4) | 58.2 (+1.0) |

**Image augmentation type.** To validate that the benefits of NoisyRollout are not limited to a single type of distortion, we evaluate its performance using rotation as a representative geometric transformation, in addition to our default Gaussian noise. The results, presented in Table 5, show that both augmentation strategies outperform the vanilla GRPO baseline. Specifically, using rotation boosts the average score from 57.2% to 58.2%. While Gaussian noise yields superior results with an average score of 59.2%, the meaningful gains from rotation demonstrate the generalizability of our approach. This suggests that the core mechanism of NoisyRollout —enhancing policy exploration through diverse visual inputs—is robust and effective across different augmentation techniques.

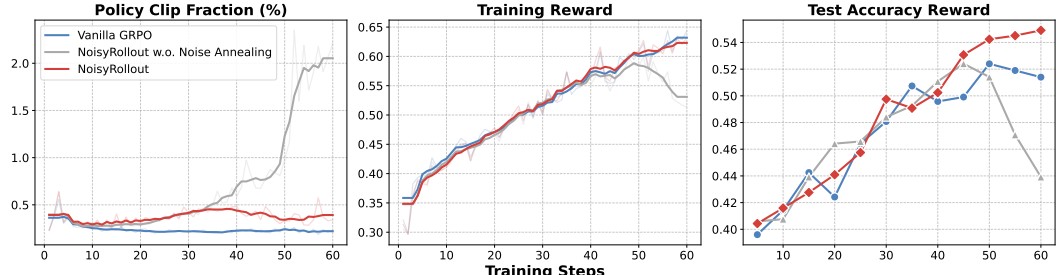

Figure 5: Comparison of NoisyRollout *w.* and *w.o.* noise annealing, and vanilla GRPO in terms of training dynamics (policy clip fraction and training reward) and accuracy on the in-domain test set.

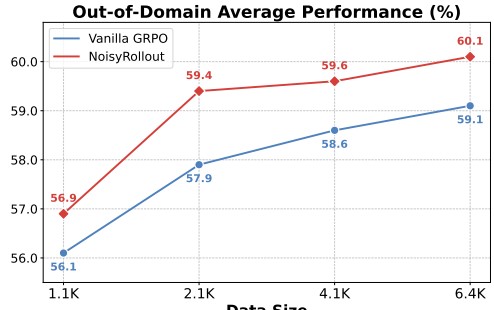

Figure 6: Performance comparison on MMK12 when scaling up the training data size.

Table 6: Ablation study on the impact of initial noise steps ($\alpha_0$). "OOD Avg." represents average accuracy (%) across five out-of-domain benchmarks.

| Noise Step | Geometry3K | OOD Avg. |
|---|---|---|
| 0 | 51.4 | 57.2 |
| 100 | 52.7 | 57.4 |
| 300 | 53.4 | 57.7 |
| 400 | 54.6 | 58.1 |
| 500 | **54.9** | **59.2** |
| 550 | 39.6 | 57.7 |
| 600 | Diverged | |

**Data scale.** Although Geometry3K is a high-quality training dataset, its limited size (2.1K samples) prevents a thorough investigation of the scaling behavior of NoisyRollout compared to vanilla GRPO. To enable such analysis, we additionally consider MMK12, which contains 6.4K samples after pre-processing. Figure 6 shows NoisyRollout consistently outperforms vanilla GRPO across various data scales, ranging from 1.1K to 6.4K. Notably, the performance gains do not diminish as the dataset size increases, suggesting that NoisyRollout has strong potential for use in large-scale training regimes.

**Initial noise step.** We evaluate the impact of noise strength by varying the initial Gaussian noise step, as shown in Table 6. Gradually increasing the initial noise step $\alpha_0$ from 0 to 500 *consistently* improves performance across all evaluation categories, suggesting that moderate noise promotes exploration and enriches the training signal. However, exceeding this threshold leads to performance degradation, as overly distorted images (see Figure 7) yield noisy rollouts with average near-zero rewards. These excessively noisy samples introduce harmful distribution shifts during policy updates, ultimately destabilizing the learning process. Additional ablation results on the MMK12 dataset are deferred to Appendix A.[7]



Figure 7: Illustration of visual degradation under increasing Gaussian noise steps.

**GRPO variant.** Recently, several variants have been proposed to enhance the original GRPO implementation. Specifically, Liu et al. [45] identified a question-level difficulty bias and proposed removing the standard deviation normalization ($\mathrm{std}(\mathbf{r})$) to address this issue. In addition, Yu et al. [87] increased the upper clipping threshold ($\epsilon_{\mathrm{high}}$) to mitigate entropy collapse. As shown in Table 7, applying NoisyRollout consistently improves performance not only on the original GRPO implementation but also across these variants. This highlights that NoisyRollout provides complementary benefits alongside optimization-focused modifications, underscoring its broad applicability.

---

[7]We also include additional ablations (e.g., **number of rollouts** and **data seed variations**) in Appendix A.

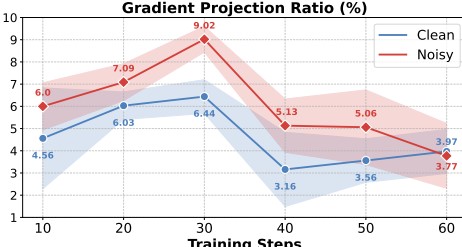

Figure 8: Comparison of gradient projection ratio over training.

Table 7: Performance comparison when using GRPO variants for policy optimization.

| Method | Geometry3K | OOD Avg. |
|---|---|---|
| Qwen2.5-VL-7B-Instruct | 39.4 | 53.3 |
| + GRPO (w.o. std($\mathbf{r}$)) | 51.3 | 57.0 |
| + NoisyRollout (w.o. std($\mathbf{r}$)) | **56.1** | 58.9 |
| + GRPO ($\epsilon_{\text{high}} = 0.28$) | 52.6 | 58.2 |
| + NoisyRollout ($\epsilon_{\text{high}} = 0.28$) | 53.9 | **59.6** |

## 3.4 Further Analysis: Quantitative Contribution of Noisy Rollouts on RL Optimization

**Setup.** For each training sample, we partition the collected rollouts into `Clean` and `Noisy` subgroups, containing $n_1$ and $n_2$ rollouts, respectively. We measure each subgroup's contribution by projecting its specific effective gradients onto an **anchor gradient** $\mathbf{g}^t = \theta^{t+\Delta t} - \theta^t$, which represents the overall model update over $\Delta t$ optimization steps, beginning at training step $t$. The subgroup effective gradients, $\mathbf{g}_{\text{clean}}^t$ and $\mathbf{g}_{\text{noisy}}^t$, are derived from actual optimization steps starting from $\theta^t$, using only rollouts from the respective subgroup (by masking losses from the other subgroup). The projection ratios are then calculated as $r_{\text{clean}}^t = (\mathbf{g}_{\text{clean}}^t \cdot \mathbf{g}^t)/\|\mathbf{g}^t\|^2$ and $r_{\text{noisy}}^t = (\mathbf{g}_{\text{noisy}}^t \cdot \mathbf{g}^t)/\|\mathbf{g}^t\|^2$. These ratios provide a quantitative estimate of each subgroup's contribution to the overall model update $\mathbf{g}^t$. More details are included in Appendix I.

**Result.** Figure 8 shows that the `Noisy` subgroup consistently contributes more significantly to policy optimization compared to `Clean`, especially during early training phases when distortion strength $\alpha_t$ is high and the policy $\pi_\theta$ still struggles with visual understanding. This trend gradually diminishes towards the final stages of training as the learning is gradually "on-policy". These findings quantitatively confirm that our method effectively leverages noisy rollouts to enhance training signals.

## 4 Related Work

VLMs have rapidly advanced through integrating vision encoders [59, 91] with large language models [2, 36, 41, 42, 24, 17], with specialized efforts in reasoning tasks [66, 94]. Reinforcement learning training in LLMs and VLMs, initially employed for alignment via human feedback (RLHF) [54, 1, 88], has evolved to incorporate rule-based rewards and advanced optimization methods like GRPO [64], as exemplified by DeepSeek-R1 [21] and Kimi-1.5 [28]. Emerging RL approaches in multimodal domains include LMM-R1 [56], Vision-R1 [23], R1-V [7], OpenVLThinker [13], and MM-Eureka [53], which extend RL to visual reasoning tasks. However, existing studies on training VLMs via RL have not adequately explored techniques that can enhance the explorative capabilities of models. Our method addresses this gap by proposing a data augmentation technique with visual-oriented inductive biases. A detailed discussion of related work is deferred to the Appendix D due to space limit.

## 5 Conclusion

In this paper, we investigate scaling test-time compute in VLMs via RL. We introduce NoisyRollout, a simple yet effective data augmentation technique that promotes diversity by mixing trajectories from both clean and distorted inputs with vision-oriented inductive biases. This approach enhances policy exploration during RL training without incurring additional training costs. Empirically, NoisyRollout demonstrates improved generalization and robustness, achieving state-of-the-art performance across multiple visual reasoning and perception benchmarks with high sample efficiency.

## Acknowledgement

This project was partially supported by the Singapore Ministry of Education Academic Research Fund Tier 1 (Award Number: T1 251RES2514) and TPU Research Cloud. Additional computational support was provided by Sea AI Lab.

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

## A Additional Ablation Studies

**Noise annealing strategy.** On the Geometry3K training dataset, we further examine the impact of different noise annealing schedules on NoisyRollout's performance by comparing our default sigmoid strategy with power ($\alpha_t = \alpha_0 \cdot (1 - t/t_{\max})^p$, $p = 3.0$) and exponential ($\alpha_t = \alpha_0 \cdot \gamma^{t/t_{\max}}$, $\gamma = 0.98$) decay functions. As shown in Table 8, all three strategies enable NoisyRollout to outperform the vanilla GRPO baseline on benchmarks. Among them, the sigmoid schedule achieves the highest average score (58.5%), surpassing both power and exponential decays (57.1% and 57.0%). The superior performance of the sigmoid schedule likely results from its characteristic "slow-fast-slow" decay, which balances exploration and stability more effectively by maintaining sufficient early exploration and promoting rapid convergence afterward.

Table 8: Ablation study on the strategies of noise annealing. "Avg." denotes the average accuracy across the six benchmarks. Best value per column is **bold**, second best is underlined.

| Method | Geo3K | MathVerse | MathVision | MathVista | WeMath | HalluBench | Avg. |
|---|---|---|---|---|---|---|---|
| Qwen2.5-VL-7B-Instruct | 39.4 | 46.2 | 25.0 | 67.5 | 63.1 | 64.6 | 51.0 |
| + GRPO | 52.0 | 50.8 | 27.3 | 70.5 | 67.4 | 69.8 | 56.3 |
| + NoisyRollout (Pow.) | 52.2 | 52.1 | 26.4 | 72.0 | 68.7 | 71.0 | 57.1 |
| + NoisyRollout (Exp.) | 51.9 | 52.6 | 27.7 | 70.5 | **70.1** | 69.1 | 57.0 |
| + NoisyRollout (Sig.) | **54.9** | **53.2** | **28.5** | **72.6** | 69.6 | **72.1** | **58.5** |

**Total rollout number.** We analyze the impact of total rollout number by comparing vanilla GRPO and NoisyRollout under varying rollout budgets. As shown in Table 9, increasing the number of rollouts in vanilla GRPO from $n = 8$ to $n = 16$ improves in-domain performance (from 49.6% to 54.7%), but only marginally benefits out-of-domain generalization (from 56.8% to 57.5%). NoisyRollout consistently outperforms vanilla GRPO even when the total number of rollouts is held constant. Notably, NoisyRollout with $n_1 = n_2 = 6$ (total 12) achieves both higher in-domain (54.9%) and out-of-domain (59.2%) accuracy than vanilla GRPO with 16 rollouts.

Table 9: Comparision of NoisyRollout and GRPO on Qwen2.5-VL-7B-Instruct across different rollout configurations. "OOD Avg." denotes the average accuracy across all five out-of-domain benchmarks.

| Rollout | Geo3K | MathVerse | MathVision | MathVista | WeMath | HalluBench | OOD Avg. |
|---|---|---|---|---|---|---|---|
| $n = 8$ | 49.6 | 50.2 | 27.3 | 68.8 | 68.1 | 69.7 | 56.8 |
| $n = 12$ | 52.0 | 50.8 | 27.3 | 70.5 | 67.4 | 69.8 | 57.2 |
| $n = 16$ | 54.7 | 51.3 | 27.5 | 71.4 | 68.0 | 69.3 | 57.5 |
| $n = 20$ | 53.6 | 50.8 | 27.2 | 70.0 | 68.5 | 69.9 | 57.3 |
| $n_1 = n_2 = 4$ | 51.7 | 51.8 | 27.4 | **72.6** | 69.5 | 70.3 | 58.3 |
| $n_1 = n_2 = 6$ | **54.9** | **53.2** | 28.5 | **72.6** | **69.6** | **72.1** | **59.2** |
| $n_1 = n_2 = 8$ | 54.7 | 52.6 | **28.7** | 72.0 | 69.1 | 72.0 | 58.9 |

**Unsuccessful image data augmentation.** In our experiments, we explored two classic image augmentation techniques: Gaussian noise and rotation with expansion (`expand=True`). Both proved effective as they introduce perceptual diversity while preserving all essential visual information. In contrast, we also investigated two augmentation strategies that were ultimately unsuccessful: cropping and rotation without expansion. The latter refers to rotating an image and then cropping it back to its original dimensions, which cuts off the corners of the rotated content. Both of these unsuccessful methods frequently resulted in **critical information loss**, as key parts of a problem or diagram were removed from the image. This led to rollouts with

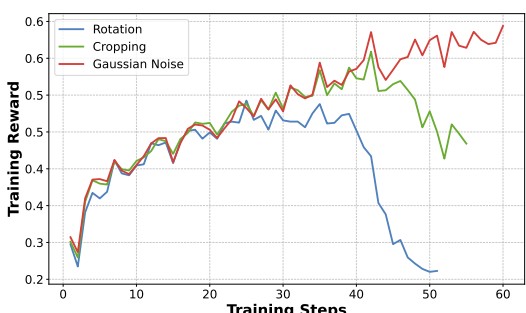

Figure 9: Performance of Rotation without expansion and cropping.

Table 10: Comparision of NoisyRollout and GRPO on Qwen2.5-VL-7B-Instruct across data seeds.

| Method | Seed | Geo3K | MathVerse | MathVision | MathVista | WeMath | HalluBench | OOD Avg. |
|---|---|---|---|---|---|---|---|---|
| GRPO | 1 | 52.0 | 50.8 | 27.3 | 70.5 | 67.4 | 69.8 | 57.2 |
| NoisyRollout | | **54.9** | 53.2 | **28.5** | 72.6 | 69.6 | **72.1** | **59.2** |
| GRPO | 2 | 51.9 | 51.5 | 27.8 | 70.7 | 68.1 | 69.4 | 57.5 |
| NoisyRollout | | 54.1 | **53.6** | 26.6 | **72.9** | 69.7 | 71.4 | 58.8 |
| GRPO | 3 | 51.1 | 50.8 | 27.1 | 69.7 | 67.8 | 69.5 | 57.0 |
| NoisyRollout | | 53.7 | 52.4 | 27.8 | 72.3 | **70.5** | 70.9 | 58.8 |
| GRPO | 42 | 51.9 | 50.5 | 27.2 | 71.3 | 68.6 | 68.6 | 57.2 |
| NoisyRollout | | 54.1 | 52.7 | 27.3 | 72.2 | 69.6 | 70.3 | 58.4 |

Table 11: Performance of NoisyRollout on the MMK12 dataset (2.1K) across different noise steps.

| Method | Initial Noise Step | MathVerse | MathVision | MathVista | WeMath | HalluBench | OOD Avg. |
|---|---|---|---|---|---|---|---|
| GRPO | 0 | 50.7 | 28.5 | 71.7 | 68.6 | 69.8 | 57.9 |
| NoisyRollout | 300 | 51.4 | 28.4 | 72.5 | 69.5 | 70.5 | 58.5 |
| | 400 | 50.3 | 28.2 | 71.8 | 70.3 | 70.8 | 58.3 |
| | 450 | **52.8** | **28.9** | 72.9 | **71.9** | 70.7 | **59.4** |
| | 500 | 50.2 | 27.4 | **73.2** | 71.4 | **71.3** | 58.7 |

consistently near-zero rewards, providing unreliable policy gradient estimates that caused training instability and eventual divergence. This distinction underscores a key finding: for NoisyRollout to be effective, the augmentation must introduce meaningful perceptual variance without fundamentally compromising the integrity of the input data.

**Data seed.** To evaluate the robustness of our approach to data sampling variations, we examine the performance consistency of NoisyRollout when using different random seeds for data sampling during training on the Geometry3K dataset. We compare these results against vanilla GRPO trained with the same seeds to verify that the improvements offered by NoisyRollout are consistent across different data orderings and not merely an artifact of a specific seed (Table 10).

**Initial noise step (MMK12).** Based on our experience, the initial noise step $\alpha_0$ is a critical hyperparameter. In addition to the Geometry3K dataset, we also evaluate the impact of different initial noise steps on the MMK12 dataset using Qwen2.5-VL-7B-Instruct. Table 11 shows NoisyRollout outperforms standard GRPO on MMK12 when configured with an appropriate initial noise step ($\alpha_0 = 450$).

**Proportion of noisy rollouts.** Table 12 demonstrates that incorporating noisy rollouts during training significantly enhances model performance across both reasoning and perception benchmarks. A balanced 50/50 distribution of clean and noisy rollouts achieves optimal results (58.5% average accuracy), outperforming both the no-noise baseline (56.3%) and higher noise proportions. This finding aligns with our earlier observations about the effectiveness of noise as a regularizer, confirming that the ideal approach combines clean rollouts for exploitation of current state with noisy rollouts for exploration, rather than relying exclusively on either strategy.

Table 12: Comparison across different proportions of noisy rollouts $n_2/(n_1 + n_2)$ during rollout collection, with a fixed total of 12 rollouts.

| Proportion | Geo3K | MathVerse | MathVision | MathVista | WeMath | HalluBench | Avg. |
|---|---|---|---|---|---|---|---|
| 0/12 | 52.0 | 50.8 | 27.3 | 70.5 | 67.4 | 69.8 | 56.3 |
| 3/12 | 51.1 | 52.8 | **28.5** | 71.8 | 69.8 | 70.6 | 57.4 |
| 6/12 | **54.9** | **53.2** | **28.5** | **72.6** | 69.6 | 72.1 | **58.5** |
| 9/12 | 54.2 | 52.1 | 27.3 | 72.3 | **71.1** | **72.3** | 57.9 |
| 12/12 | 53.1 | 52.8 | **28.5** | 72.3 | 69.1 | 71.1 | 57.8 |

# B Unsuccessful Attempts

During the early development of NoisyRollout, we encountered several unsuccessful trials that are worth documenting. Due to limited computational resources, we could only explore a limited set of hyperparameter combinations heuristically. Some of these approaches might prove effective with further hyperparameter optimization and better design.

**Optimizing noisy and clean trajectories on corresponding inputs.** We explored an alternative design in which policy updates were conditioned on the same input used to generate each rollout—i.e., using clean inputs for clean rollouts and distorted inputs for noisy rollouts. However, this approach did not yield meaningful improvements over the original GRPO baseline. We hypothesize that this design reduces the benefits of group-based advantage estimation. Specifically, by decoupling clean and noisy rollouts during optimization, the method effectively degrades into a form of sample-level data augmentation. This fragmentation weakens the shared reward signal across rollouts, thereby diminishing the informativeness of group-level statistics such as the normalized advantage. As a result, the exploration benefits introduced by noisy rollouts are not fully leveraged during policy updates.

In contrast, our proposed approach treats clean and noisy rollouts as a unified group for advantage calculation, while anchoring all policy optimization to the clean inputs. This design retains the distributional diversity introduced by noise, but preserves a consistent input distribution for policy updates—striking a balance between exploration and stable learning.

**Reward penalty on noisy subgroup.** We experimented with applying explicit reward penalties (e.g., $-0.1$) to all noisy rollouts, aiming to encourage the model to better capture contrastive learning signals. However, this approach quickly led to training divergence. Rather than improving its core reasoning and perception abilities, the policy model learned to distinguish between clean and noisy rollouts. As a result, the noisy rollouts rapidly became highly "off-policy", since the model could easily identify. This distributional mismatch destabilized training and undermined the learning.

# C  Evaluating the Perception Quality of Reasoning Traces

---

**Visual Information Extraction Prompt**

Extract all visual perception and information recognition components from the following reasoning trace.
Original question: {question}
Reasoning trace: {reasoning}

Your task is to extract and summarize ONLY the parts that relate to visual perception, information extraction, and understanding of visual elements from the image.

This includes:
1. Any measurements, dimensions, or numerical values extracted from the image
2. Description of visual elements like shapes, objects, positions, or spatial relationships
3. Recognition of text, symbols, diagrams, or graphs from the image
4. Any visual features mentioned or used in the reasoning

Format your response with the tag:
<visual_perception> [Extracted visual information here] </visual_perception>

Include ONLY visual perception elements, not mathematical reasoning that happens after the information is extracted. If there are no clear visual perception elements, respond with "No clear visual perception elements identified."

**Visual Perception Comparison Prompt**

Compare the quality of visual perception between two models based on the image and the original question.
Original question: {question}
Visual perception from Model A: {visual A}
Visual perception from Model B: {visual B}

Your task is to determine which model better captures and correctly extracts visual information from the image. Compare their visual perception quality based on:

- Accuracy of visual information extraction (measurements, shapes, relationships)

- Complete identification of all relevant visual elements

- Proper recognition of visual information required to solve the problem

Score both models and determine the winner: If Model A demonstrates significantly better perception than Model B, respond: <result>A</result>; if Model B demonstrates significantly better perception than Model A, respond: <result>B</result>; if both models show similar quality of visual perception, respond: <result>tie</result>.

Now:
1. identify what visual information is required to solve this problem.
2. analyze how each model perceives this information.
3. provide your comparative judgment with specific reasons.
4. provide your <result> tag with exactly A, B, or tie.

---

To further evaluate the perception quality of models trained with NoisyRollout and vanilla GRPO during reasoning, we perform a paired comparison using a strong VLM.[8] We sample 300 reasoning traces from the evaluation logs of the models performing visual reasoning on the MathVerse and MathVista benchmarks, forming paired comparisons between NoisyRollout and vanilla GRPO.

To isolate visual perception, we extract only the visual components from each reasoning trace using a specialized prompt, removing any influence from mathematical reasoning or final answers. To reduce potential position bias in the comparisons, each pair of traces is evaluated twice: once with the

---

[8]Specifically, we use Gemini-Flash-2.0-001 in this experiment.

NoisyRollout trace shown first and the vanilla GRPO trace second, and once with the order reversed. We combine the results using the Bradley-Terry model to compute win rates. This methodology offers a reliable measure focused specifically on visual perception quality during reasoning. The results are presented in Figure 3 (the 8th subfigure). The extraction and evaluation prompts are shown above.

# D  Detailed Related Work

**Large Vision-Language Models.**  VLMs have rapidly evolved to understand and reason with both visual and textual information [70]. These models combine visual encoders with large language models to enable comprehension and inference across modalities. Early VLMs like Flamingo [2] and BLIP-2 [36] established foundational integration techniques between vision and language components. The LLaVA series [41, 42, 35, 34] introduced effective visual instruction tuning methodologies that significantly advanced multimodal capabilities. For mathematical reasoning, specialized approaches [66, 94] have employed mathematical visual instruction tuning to enhance VLMs' abilities to interpret and solve mathematical problems in multimodal contexts.

Advanced VLMs including GPT-4o [24] and Gemini [17] have demonstrated unprecedented general visual understanding through massive pretraining. Mixture-of-Experts approaches in DeepSeek-VL-2 [80], Uni-MoE [38], and MoVA [98] improved computational efficiency by selectively activating specialized components based on input characteristics. Meanwhile, unified models like SEED-X [16], Chameleon [69], Show-o [81], and Janus series [79, 52, 8] integrated visual understanding and generation capabilities within single architectures. However, most existing VLMs still lack robust visual reasoning capabilities [14], especially for tasks requiring sophisticated analysis of visual information combined with complex reasoning [43, 75].

**Reinforcement Learning-Enhanced Visual Reasoning.**  RL has emerged as a key methodology for enhancing the capabilities of LLMs and VLMs. Early research primarily focused on Reinforcement Learning from Human Feedback (RLHF) [54], which aligned model outputs with human preferences [1]. Recent advancements have further demonstrated that RL-based techniques can significantly enhance reasoning abilities. For instance, DeepSeek-R1 [21] utilizes rule-based rewards combined with Group Relative Policy Optimization (GRPO) [64], whereas Kimi-1.5 [28] employs a variant of online policy mirror descent, both methods showing notable improvements in reasoning performance.

In the multimodal domain, research on leveraging RL to enhance VLMs' reasoning capabilities remains in early stages. Some approaches explore using generative reward models [95, 76] to enhance VLMs' general capability, but these typically require powerful closed-source models for training data generation. Recent work including LMM-R1 [56], Vision-R1 [23], R1-V [7] and OpenVLThinker [13] has applied R1-type RL to VLMs in diverse specific subdomains like geometry problems and object counting tasks [55, 10, 12, 39, 37, 73, 26]. Further more, pilot studies [53, 51] further extend large-scale rule-based RL to broader multimodal mathematical reasoning, demonstrating significant performance gains without relying on in-domain training data.

**Data Augmentation in Visual Reinforcement Learning.**  Data augmentation has played a central role in visual reinforcement learning (RL) from pixels by improving sample efficiency and generalization without architectural changes. RAD (Reinforcement Learning with Augmented Data) first demonstrated that simple image transformations—random crop, translation, color jitter, and cutout—can substantially enhance policy learning, particularly in DMControl and ProcGen benchmarks [33]. CURL combined off-policy RL with contrastive learning on augmented views, improving representation quality and sample efficiency [68]. DrQ regularized Q-value targets and policy updates via random shifts, achieving stable gains with minimal computational overhead [30]; DrQ-v2 further refined training schedules, exploration, and multi-step targets to master high-dimensional control from pixels [85].

Beyond these baselines, several extensions explored stronger invariance and generalization. DrAC introduced actor–critic regularizers that promote consistency under augmentation and automated augmentation search for diverse environments [60]. SVEA mitigated instability under heavy augmentations by mixing augmented and clean views to reduce target variance [22]. Self-Predictive Representations (SPR) encouraged temporal consistency between augmented latent states to improve representation quality and low-data efficiency [63]. In the sim-to-real context, domain randomization perturbs textures, lighting, and geometry to improve transferability from simulation to real-world environments [71, 61].

While these studies have established augmentation as a key ingredient for robust visual control, most methods optimize control policies rather than multimodal reasoning. Current augmentation pipelines are designed to stabilize perception and dynamics modeling but rarely connect augmented visual representations to higher-level symbolic or linguistic reasoning. Bridging this gap will likely require augmentations that preserve both control-relevant and semantic information, enabling joint optimization for perception, reasoning, and alignment in future multimodal agents.

## E    Broader Impacts

Our proposed method, **NoisyRollout**, introduces a simple yet powerful data augmentation approach designed to improve visual reasoning and perceptual robustness in VLMs. Given its effectiveness, especially noted through strong out-of-domain performance and high sample efficiency, this approach has broad applicability within resource-constrained training scenarios. This is particularly beneficial in domains where acquiring or annotating large-scale datasets is costly or practically challenging, such as medical imaging [32], robotic perception [67], and assistive technologies [15].

Moreover, by enhancing model robustness to visual conditions, our method can also facilitate safer and more reliable deployment of VLMs in real-world applications, potentially leading to more trustworthy human-AI interactions. Furthermore, as our method involves relatively simple augmentation steps without additional computational overhead or complex training protocols, along with strong performance on scaling experiments as shown in Table 2 and Figure 6, it is suitable for integration into existing large-scale training pipelines, supporting broader adoption in both academia and industry.

## F  Limitations

Despite easy-to-adopt designs and promising empirical results, our study has several limitations. **First**, due to computational constraints, our experiments are limited in scale: we primarily explore model sizes up to 32B parameters and training dataset scales in the order of a few thousand samples. Future work should validate and extend our findings using significantly larger-scale training scenarios—such as models with 72B parameters or training datasets in the range of hundreds of thousands of sample. **Second**, NoisyRollout is applied during the RL fine-tuning phase of an already pre-trained VLM. A more fundamental, but vastly more complex, direction would be to explore how the principles of NoisyRollout (i.e., learning from noisy signals in RL) could be integrated into the large-scale pre-training phase of the VLM itself. **Finally**, while empirically effective, our study lacks a formal theoretical analysis of how NoisyRollout, with its specific hybrid trajectory mixing and noise annealing, affects the exploration-exploitation trade-off and the convergence properties of the RL algorithm. It's unclear if the introduced noise guarantees broader state-space coverage in a principled way or if certain noise characteristics could inadvertently hinder convergence.

## G  Case Study

We present two case studies to demonstrate the improved perception and reasoning capabilities of our NoisyRollout compared to vanilla GRPO, as illustrated in Figure 10 and Figure 11, respectively.

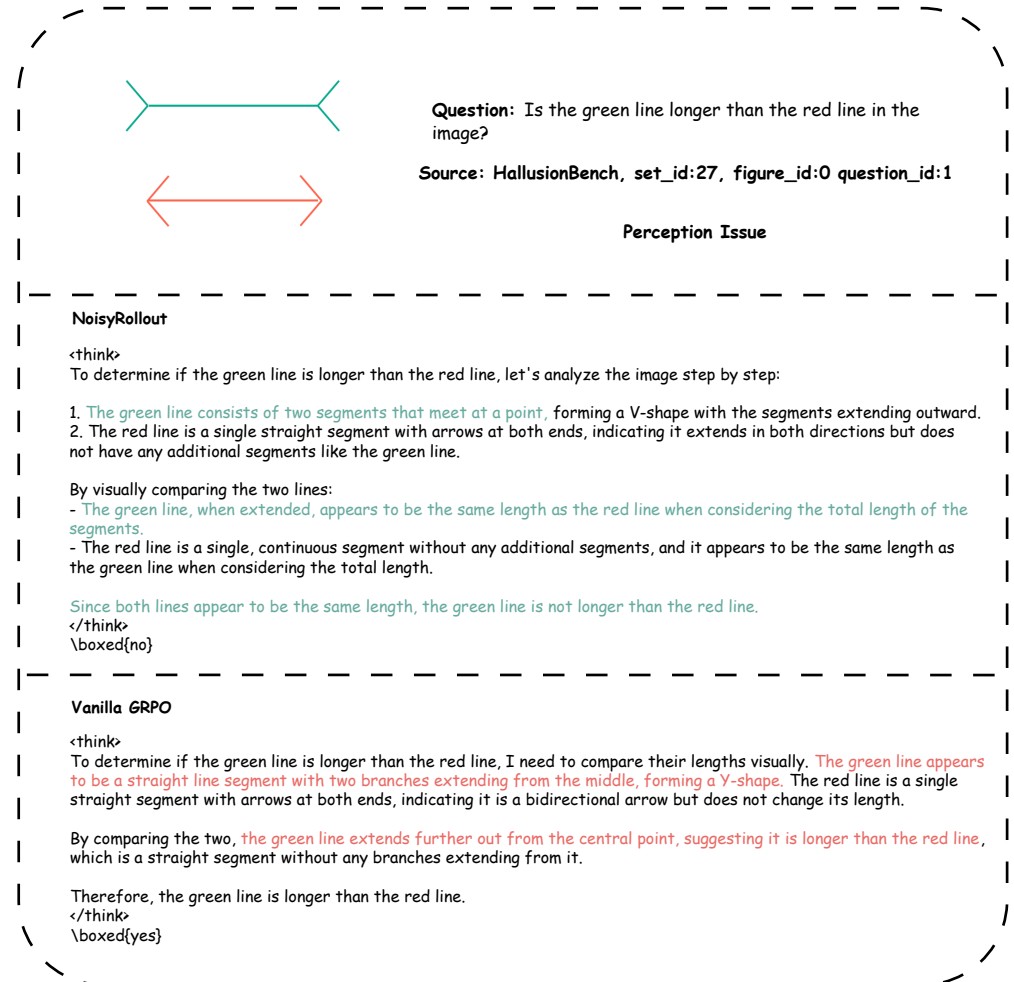

Figure 10: Case study showing improved perception capability of NoisyRollout over vanilla GRPO.

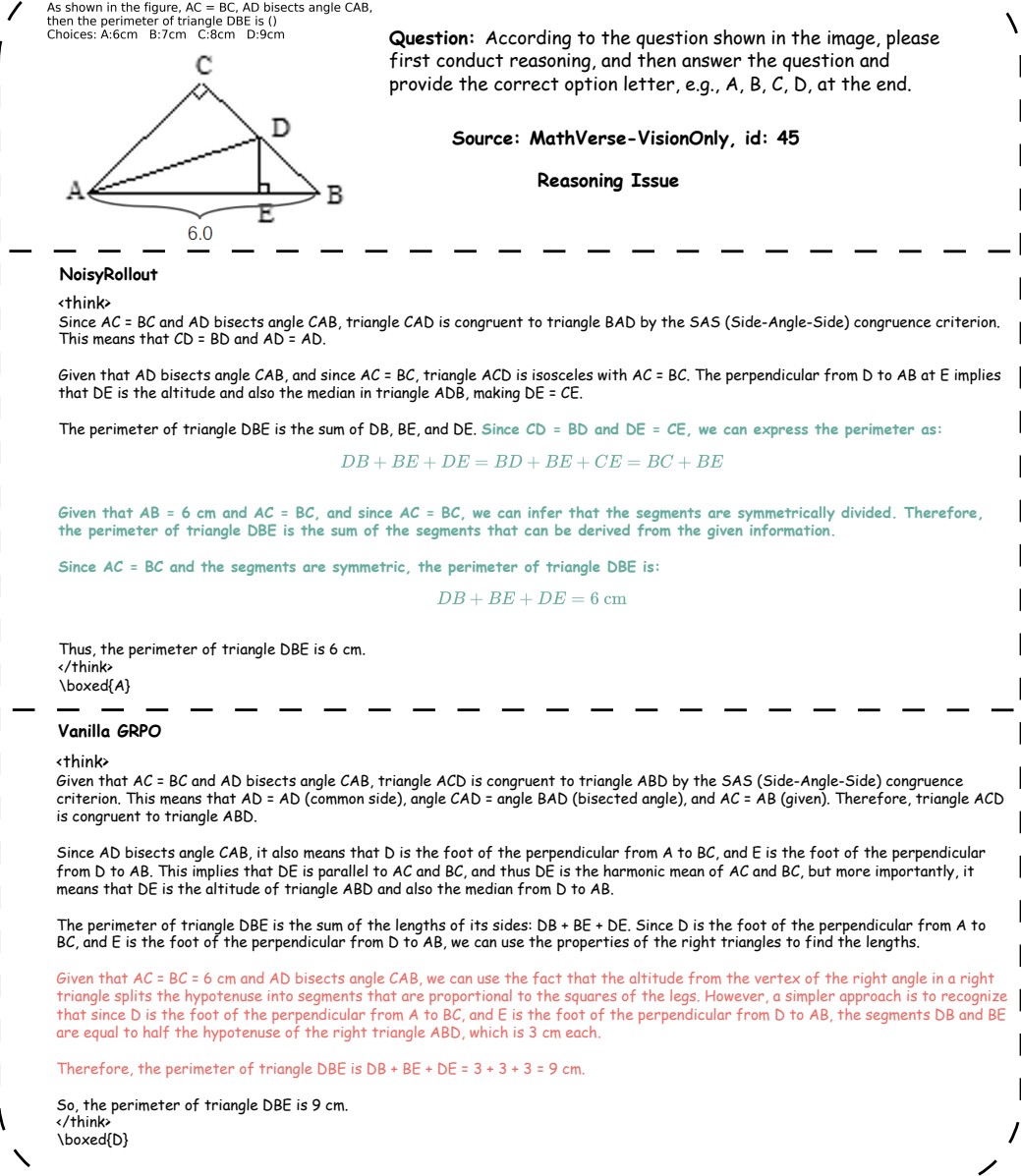

As shown in the figure, AC = BC, AD bisects angle CAB, then the perimeter of triangle DBE is ()
Choices: A:6cm   B:7cm   C:8cm   D:9cm

**Question:** According to the question shown in the image, please first conduct reasoning, and then answer the question and provide the correct option letter, e.g., A, B, C, D, at the end.

**Source: MathVerse-VisionOnly, id: 45**

**Reasoning Issue**

**NoisyRollout**

<think>
Since AC = BC and AD bisects angle CAB, triangle CAD is congruent to triangle BAD by the SAS (Side-Angle-Side) congruence criterion. This means that CD = BD and AD = AD.

Given that AD bisects angle CAB, and since AC = BC, triangle ACD is isosceles with AC = BC. The perpendicular from D to AB at E implies that DE is the altitude and also the median in triangle ADB, making DE = CE.

The perimeter of triangle DBE is the sum of DB, BE, and DE. Since CD = BD and DE = CE, we can express the perimeter as:

$$DB + BE + DE = BD + BE + CE = BC + BE$$

Given that AB = 6 cm and AC = BC, and since AC = BC, we can infer that the segments are symmetrically divided. Therefore, the perimeter of triangle DBE is the sum of the segments that can be derived from the given information.

Since AC = BC and the segments are symmetric, the perimeter of triangle DBE is:

$$DB + BE + DE = 6 \text{ cm}$$

Thus, the perimeter of triangle DBE is 6 cm.
</think>
\boxed{A}

**Vanilla GRPO**

<think>
Given that AC = BC and AD bisects angle CAB, triangle ACD is congruent to triangle ABD by the SAS (Side-Angle-Side) congruence criterion. This means that AD = AD (common side), angle CAD = angle BAD (bisected angle), and AC = AB (given). Therefore, triangle ACD is congruent to triangle ABD.

Since AD bisects angle CAB, it also means that D is the foot of the perpendicular from A to BC, and E is the foot of the perpendicular from D to AB. This implies that DE is parallel to AC and BC, and thus DE is the harmonic mean of AC and BC, but more importantly, it means that DE is the altitude of triangle ABD and also the median from D to AB.

The perimeter of triangle DBE is the sum of the lengths of its sides: DB + BE + DE. Since D is the foot of the perpendicular from A to BC, and E is the foot of the perpendicular from D to AB, we can use the properties of the right triangles to find the lengths.

Given that AC = BC = 6 cm and AD bisects angle CAB, we can use the fact that the altitude from the vertex of the right angle in a right triangle splits the hypotenuse into segments that are proportional to the squares of the legs. However, a simpler approach is to recognize that since D is the foot of the perpendicular from A to BC, and E is the foot of the perpendicular from D to AB, the segments DB and BE are equal to half the hypotenuse of the right triangle ABD, which is 3 cm each.

Therefore, the perimeter of triangle DBE is DB + BE + DE = 3 + 3 + 3 = 9 cm.

So, the perimeter of triangle DBE is 9 cm.
</think>
\boxed{D}

Figure 11: Case study illustrating enhanced reasoning capability of NoisyRollout over vanilla GRPO.

## H    Templates

**Reasoning Template from EasyR1**

**SYSTEM:** You FIRST think about the reasoning process as an internal monologue and then provide the final answer.The reasoning process MUST BE enclosed within <think> </think> tags. The final answer MUST BE put in \boxed{}.
**USER:** {question}

**Direct-Answer Template**

**SYSTEM:** You are a helpful assistant
**USER:** {question}. Answer yes or no directly.

# I  Detailed Methodology for Gradient Contribution Analysis

To quantitatively assess the impact of noisy rollouts on the reinforcement learning (RL) optimization process, we partition the rollouts associated with each training sample into two distinct subgroups based on their input type: `Clean` and `Noisy`. Specifically, `Clean` rollouts are generated from original inputs $(I, \mathbf{q})$, whereas `Noisy` rollouts originate from distorted inputs $(\tilde{I}, \mathbf{q})$. In this experimental setup, each training sample comprises $n_1 = 6$ `Clean` rollouts and $n_2 = 6$ `Noisy` rollouts. For computational efficiency, all gradient calculations and parameter differences discussed below ($\mathbf{g}^t$, $\mathbf{g}^t_{\text{clean}}$, $\mathbf{g}^t_{\text{noisy}}$) are performed using only the parameters from specific model modules. The exact modules used are "`lm_head.weight`", "`model.layers.27.self_attn.o_proj`", "`model.layers.27.self_attn.q_proj`", "`model.layers.27.self_attn.k_proj`", and "`model.layers.27.self_attn.v_proj`".

To quantify the contribution of each subgroup to the optimization, we first define an **anchor gradient**, denoted $\mathbf{g}^t$. This quantity represents the effective overall update to the selected model parameters $\theta$ at a given training stage $t$. It is calculated as the difference in these parameters between checkpoints at training steps $t$ and $t + \Delta t$:

$$\mathbf{g}^t = \theta^{t+\Delta t} - \theta^t,$$

where $\theta^t$ represents the selected model parameters at training step $t$, and we use $\Delta t = 5$ steps. This $\mathbf{g}^t$ reflects the actual change in these parameters resulting from the standard training procedure which utilizes losses derived from both `Clean` and `Noisy` rollouts from each training sample.

Subsequently, starting from the same parameter state $\theta^t$, we isolate the influence of each subgroup. This involves performing $\Delta t$ actual optimization steps under two modified conditions, using the same batch of training samples that contributed to the standard update from $\theta^t$ to $\theta^{t+\Delta t}$:

1. To obtain $\theta^{t+\Delta t}_{\text{clean}}$: Starting from $\theta^t$, we performed $\Delta t$ optimization steps. During these steps, the loss components arising from the `Noisy` rollouts were masked (i.e., their corresponding loss was set to zero). Thus, the gradients and subsequent parameter updates were derived solely from the `Clean` rollouts. This procedure yielded the updated parameters $\theta^{t+\Delta t}_{\text{clean}}$.

2. To obtain $\theta^{t+\Delta t}_{\text{noisy}}$: Similarly, starting from $\theta^t$, we performed $\Delta t$ optimization steps. In this case, the loss components arising from the `Clean` rollouts were masked. The gradients and subsequent parameter updates were therefore derived solely from the `Noisy` rollouts. This yielded updated parameters $\theta^{t+\Delta t}_{\text{noisy}}$.

From these updates, we define the subgroup-specific effective gradients (or parameter deltas) with respect to the selected modules:

$$\mathbf{g}^t_{\text{clean}} = \theta^{t+\Delta t}_{\text{clean}} - \theta^t$$

and

$$\mathbf{g}^t_{\text{noisy}} = \theta^{t+\Delta t}_{\text{noisy}} - \theta^t.$$

We then quantify the contribution of each subgroup by projecting its effective gradient onto the anchor gradient. The projection ratios are computed as follows:

$$r^t_{\text{clean}} = \frac{\mathbf{g}^t_{\text{clean}} \cdot \mathbf{g}^t}{\|\mathbf{g}^t\|^2} \quad \text{and} \quad r^t_{\text{noisy}} = \frac{\mathbf{g}^t_{\text{noisy}} \cdot \mathbf{g}^t}{\|\mathbf{g}^t\|^2}.$$

These ratios, $r^t_{\text{clean}}$ and $r^t_{\text{noisy}}$, represent the estimated proportion of the anchor gradient $\mathbf{g}^t$ that can be attributed to the `Clean` and `Noisy` subgroups, respectively, at training stage $t$. To ensure robust estimates, all effective gradient quantities ($\mathbf{g}^t$, $\mathbf{g}^t_{\text{clean}}$, and $\mathbf{g}^t_{\text{noisy}}$) are determined by averaging results from 5 independent runs of the $\Delta t$-step update processes described above. Each run starts with the same parameters $\theta^t$.

## J    Supplementary Implementation Details

This section provides the detailed hyperparameter configurations for our experiments that were omitted from Section 3. In Table 13, we summarize our experimental settings across different model sizes and datasets, with specific focus on image distortion parameters and noise annealing schedules.

Table 13: Summary of hyperparameter configurations.

| Parameter | Configuration |
|---|---|
| **General Settings (All Experiments)** | |
| Model Base | Qwen2.5-VL-Instruct |
| Vision Encoder | Frozen |
| Global Batch Size | 128 |
| Rollout Batch Size | 512 |
| Rollout Temperature | 1.0 |
| Learning Rate | $1e-6$ |
| Optimizer | AdamW |
| Policy Loss Aggregation | `token-mean` |
| Image Distortion Strategy | Gaussian noise |
| Noise Annealing Schedule | Sigmoid-shaped |
| CPU Memory | 1TB |
| GPU | A100-SXM4-40/80GB |
| **Qwen2.5-7B-VL-Instruct on Geometry3K (2.1K samples)** | |
| Initial Noise ($\alpha_0$) | 500 |
| Training Episodes | 15 |
| Total Optimization Steps ($t_{\max}$) | 60 |
| Sigmoid Midpoint ($\gamma$) | 40 |
| Sigmoid Steepness ($\lambda$) | 30 |
| Rollout Number | $n_1 = n_2 = 6$ |
| Time Cost per Step | about $1100s$ |
| **Qwen2.5-7B-VL-Instruct on MMK12 (6.4K samples)** | |
| Initial Noise ($\alpha_0$) | 450 |
| Training Episodes | 12 |
| Total Optimization Steps ($t_{\max}$) | 120 |
| Sigmoid Midpoint ($\gamma$) | 40 |
| Sigmoid Steepness ($\lambda$) | 60 |
| Rollout Number | $n_1 = n_2 = 6$ |
| Time Cost per Step | about $1500s$ |
| **Qwen2.5-32B-VL-Instruct on Geometry3K (2.1K samples)** | |
| Initial Noise ($\alpha_0$) | 450 |
| Training Episodes | 10 |
| Total Optimization Steps ($t_{\max}$) | 40 |
| Sigmoid Midpoint ($\gamma$) | 35 |
| Sigmoid Steepness ($\lambda$) | 30 |
| Rollout Number | $n_1 = n_2 = 4$ |
| Time Cost per Step | about $3300s$ |
| **Qwen2.5-32B-VL-Instruct on MMK12 (6.4K samples)** | |
| Initial Noise ($\alpha_0$) | 450 |
| Training Episodes | 7 |
| Total Optimization Steps ($t_{\max}$) | 70 |
| Sigmoid Midpoint ($\gamma$) | 35 |
| Sigmoid Steepness ($\lambda$) | 30 |
| Rollout Number | $n_1 = n_2 = 4$ |
| Time Cost per Step | about $3300s$ |

# K  Licenses

We use standard licenses from the community. We include the following licenses for the codes, datasets and models we used in this paper.

Datasets & Benchmarks:

- Geometry3K [47]: MIT
- MMK12 [53]: Apache License 2.0
- MathVerse [93]: MIT
- MathVision [74]: MIT
- MathVista [48]: Creative Commons Attribution Share Alike 4.0 International
- WeMath [57]: CC BY-NC 4.0

Codes:

- verl [65]: Apache License 2.0
- EasyR1 [96]: Apache License 2.0

Models:

- Qwen2.5-VL-7B-Instruct [4]: Apache License 2.0
- Qwen2.5-VL-32B-Instruct [4]: Apache License 2.0
- Gemini API [18]: Gemini API Additional Terms of Service

