# OpenReview forum: "NoisyRollout: Reinforcing Visual Reasoning with Data Augmentation"
_NeurIPS.cc/2025/Conference — NeurIPS 2025 poster_

### Official Review · Reviewer_5RUf · 2025-06-12

**Clarity:** 2
**Significance:** 2
**Originality:** 2
**Rating:** 4
**Confidence:** 3

**Summary:**

The paper proposes NoisyRollout, a reinforcement learning (RL) framework that improves vision-language models (VLMs) on reasoning tasks.
The key idea is to augment rollout diversity by injecting noise into training images.

Specifically, for each image–text query pair, the algorithm performs the following steps:
- roll out policies on both clean and distorted images;
- compute normalized advantages under Group Relative Policy Optimization (GRPO);
- perform policy updates only on clean inputs.

Experiments on five benchmarks (MathVerse, MathVision, MathVista, WeMath, HallusionBench) show that NoisyRollout improves over vanilla GRPO and other RL-tuned VLMs.

**Questions:**

I have questions about novelty and contributions in the section above.

Regarding the algorithm, could the authors clarify that if the updates are only performed on clean inputs, how does the model learn to reason on imperfect images?
If I understand correctly, the rewards on the distorted inputs only affect the baseline in the advantage computation.

**Ethical Concerns:**

["NO or VERY MINOR ethics concerns only"]

**Final Justification:**

The authors' clarifications have addressed my initial misunderstandings and concerns. While the proposed algorithm is not fundamentally novel compared with existing sample-level augmentation approaches, it is simple and shows clear empirical improvements.

**Limitations:**

The limitations are discussed in the appendix.

**Quality:**

2

**Strengths And Weaknesses:**

### Strengths

The proposed hybrid rollout scheme is simple. It's compatible with GRPO and requires no extra training cost (other than augmenting the training set with noisy images).

Empirically, the proposed method outperforms several baselines.

### Weaknesses

**Novelty and contributions.**
The contribution focuses on the robustness of a reasoning VLM on imperfect images. The key idea is to inject noise into training images, which is a standard data augmentation technique.
Although the paper claims to improve visual reasoning, the improvements appear to benefit more from robust perception than from improving the reasoning ability.

---

> ### Author Rebuttal · Authors · 2025-07-31
>
> Thank you for your valuable review and suggestions. Below we respond to the comments in **Weaknesses (W)** and **Questions (Q)**.
> > ***W1.1: The key idea is to inject noise into training images, which is a standard data augmentation technique.***
> >
>
> ### **Generalizability Beyond Gaussian Noise**
>
> We previously noted that other augmentation strategies (e.g., rotation, cropping) can easily lead to critical information loss, thereby making the rollouts meaningless or even harmful for policy optimization. However, after carefully reviewing the image augmentation relevant code in our codebase, we found that our rotation implementation used suboptimal parameters (`expand=False`) that caused information loss. After adopting more appropriate parameters (`image.rotate(max_angle, resample=Image.BILINEAR, expand=True)`), rotation now demonstrates meaningful improvements:
>
> | Method | Image Augmentation | MathVerse | MathVision | MathVista | WeMath | HallusionBench | Average |
> | --- | --- | --- | --- | --- | --- | --- | --- |
> | GRPO | None | 50.8 | 27.3 | 70.5 | 67.4 | 69.8 | 57.2 |
> | GRPO | Gaussian noise | **53.2** | **28.5** | **72.6** | **69.6** | **72.1** | **59.2** |
> | GRPO | Rotation | 52.5 | 28.1 | 71.9 | 68.1 | 70.2 | 58.2 |
>
> This indicates that our NoisyRollout framework can work with various augmentation strategies which would not cause visual information loss. While Gaussian noise remains most effective, likely because it preserves all visual content while adding controlled perturbations, we argue other augmentations can also provide benefits when properly implemented.
>
> ### **Rollout-Level vs. Sample-Level Augmentation**
>
> While noise injection is a common data augmentation technique, we respectfully argue that our NoisyRollout is not a straightforward adaptation of standard data augmentation in terms of its specific implementation. Below we show the fundamental differences between traditional implementation and our specific design for reinforcement learning with verifiable rewards (RLVR) + VLM:
>
> - **Traditional sample-level augmentation (DrQ [2], RAD [3])**: Rollouts are collected from both original and augmented images, with each trajectory optimized according to its actual source image during policy updates.
> - **Our rollout-level augmentation (NoisyRollout)**: We collect trajectories from both original and augmented images, but during policy optimization, we treat all rollouts as if they originated from clean images, regardless of their actual source, as indicated in Equation (2).
>
> In our appendix (lines 616-624), we have previously noted that applying augmentation at the sample level is less effective than our proposed rollout-level augmentation, and as part of our rebuttal, we provide additional quantitative experiments to substantiate this claim:
>
> - **Setup**: Trained on the Geometry3K dataset (2.1K samples) with Qwen2.5-7B-VL-Instruct; `rollout=6+6` denotes 6 rollouts from clean images and 6 rollouts from noisy images
> - **Observation**: Rollout-level augmentation consistently outperforms both baseline and sample-level augmentation across all benchmarks
> - **Why it works**: Sample-level augmentation restricts exploration since augmented samples stay tied to their original image-question pairs ("on-policy"). Rollout-level augmentation enables diverse exploration through visual-oriented inductive bias, helping models benefit from more effective supervision signals brought by noisy rollouts. A recent study in the LLM domain [1] shares a similar observation regarding the effectiveness of "off-policy" rollouts.
>
> | Method | Augmentation Type | MathVerse | MathVision | MathVista | WeMath | HallusionBench | Average |
> | --- | --- | --- | --- | --- | --- | --- | --- |
> | GRPO | None (rollout=12) | 50.8 | 27.3 | 70.5 | 67.4 | 69.8 | 57.2 |
> | GRPO | Sample-level (rollout=6+6) | 49.5 | 28.4 | 69.3 | 67.4 | 68.6 | 56.6 |
> | GRPO | Sample-level (rollout=12+12) | 50.7 | 26.5 | 70.1 | 67.9 | 70.0 | 57.0 |
> | GRPO | Rollout-level (rollout=6+6) | **53.2** | **28.5** | **72.6** | **69.6** | **72.1** | **59.2** |
>
> ***Note**: The above table shows sample-level augmentation underperforms vanilla GRPO. Unlike standard sample-level methods that randomly sample augmented data, we intentionally included both original and augmented versions of the same image within each batch to maintain consistency with our rollout-level approach (which includes all rollouts from a given sample in one batch). This design choice may hurt sample-level performance, though optimizing the sampling strategy is left for future work.*
>
> In summary, NoisyRollout introduces a simple, effective rollout-level augmentation strategy that specifically addresses visual reasoning challenges through vision-oriented inductive biases in the emerging RLVR+VLM domain.
>
> [1]. Yan, Jianhao, et al. "Learning to reason under off-policy guidance." (arXiv 2025)
>
> [2]. Yarats, Denis, Ilya Kostrikov, and Rob Fergus. "Image augmentation is all you need: Regularizing deep reinforcement learning from pixels." (ICLR 2021)
>
> [3]. Laskin, Misha, et al. "Reinforcement learning with augmented data." (NeurIPS 2020)
>
> ---
>
> > ***W1.2: Although the paper claims to improve visual reasoning, the improvements appear to benefit more from robust perception than from improving the reasoning ability.***
> >
>
> GRPO explicitly optimizes accuracy, making it inherently challenging to separate gains in reasoning from perception enhancements.  Moreover, within VLM contexts, robust perception naturally supports better reasoning, and improved reasoning, in turn, guides the model's visual focus. Hence, distinguishing strictly between perception and reasoning improvements is less meaningful than recognizing their combined contribution to overall performance in our this context.
>
> However, we acknowledge that clearly defining and quantifying the interaction between perception and reasoning merits further exploration in the domain of Explainable AI (XAI). We leave this as our future work.
>
> ---
>
> > ***Q1.1: Could the authors clarify that if the updates are only performed on clean inputs, how does the model learn to reason on imperfect images?***
> >
>
> Thank you for this clarification question. You're correct that policy updates are conditioned on clean inputs. Regarding how the model learns robust visual reasoning on imperfect images:
>
> - Noisy rollouts sampled from distorted images simulate challenging training conditions where the model must solve tasks despite imperfect visual perception;
> - Successful noisy rollouts usually offer more diverse solutions than traditional rollouts, while failed attempts (rollouts with negative rewards) teach the model to avoid perceptual reasoning patterns that are difficult to learn through the normal exploration.
>
> ---
>
> > ***Q1.2: If I understand correctly, the rewards on the distorted inputs only affect the baseline in the advantage computation.***
> >
>
> We want to clarify that the rewards on the distorted inputs do **NOT** only affect the baseline in the advantage computation. Specifically, we treat noisy rollouts equivalently to those generated from clean images during policy optimization. The noisy rollouts not only contribute to reward baseline and advantage computation, **but are also rewarded and used for policy updates (conditioned on clean images).** This provides a special form of "off-policy" exploration that increases rollout diversity, enabling the policy to explore more effectively in the solution space (or environment).

---

> > ### Author Response · Authors · 2025-08-05
> >
> > Dear Reviewer 5RUf,
> >
> > Thank you for your valuable feedback on our manuscript. Upon your request, in our previous response, we have:
> >
> > - Addressed novelty concerns by demonstrating our rollout-level augmentation differs fundamentally from sample-level approaches with quantitative evidence.
> >
> > - Demonstrated generalizability beyond Gaussian noise by showing our framework works with various augmentation strategies.
> >
> > - Explained how the model learns on imperfect images through noisy rollouts that provide diverse supervision signals.
> >
> > - Clarified that noisy rollouts contribute to both reward computation and policy updates, not just baseline computation.
> >
> >
> >
> > We would appreciate it if you could let us know if our response has addressed your concerns, and we kindly request a reconsideration of the score.
> >
> > Best,
> >
> > Authors

---

> ### Comment · Reviewer_5RUf · 2025-08-07
>
> Thanks for the clarifications and the additional results. The algorithm is clearer to me: It collects rollouts from original inputs and noisy inputs, and uses them (and their rewards) to update the policy for the original inputs.
>
> I also find the following to be insightful in explaining why this algorithm works. It would be great to highlight this in the paper.
> > Successful noisy rollouts usually offer more diverse solutions than traditional rollouts.
>
> I still have a question about what the objective is. I think there are two possibilities:
> 1) We only care about the performance on the original inputs, and assume the test-time inputs are clear.
> 2) We care about the performance on both original and noisy inputs, that is, we want the model to be visually robust.
>
> For 1), the algorithm in the paper is appropriate. We collect more diverse rollouts using noisy input, and then use all rollouts to learn $\pi(\cdot | I, q)$.
>
> For 2), we would want to learn both $\pi(\cdot | I, q)$ and $\pi(\cdot | \tilde{I}, q)$, since we care about the model's performance on noisy inputs as well?

---

> ### Author Response · Authors · 2025-08-07
> **Thank you for your response!**
>
> Thank you for your response. Yes, exactly—your understanding is right on point! We’re glad our clarification was helpful. We actually considered both possibilities you mentioned at the very start of this project. As we discussed before, these can be roughly divided into rollout-level augmentation (ours) and sample-level augmentation.
>
> While the second possibility you brought up might help make models a bit more visually robust, our experiments showed that it’s not as effective as NoisyRollout on visual reasoning benchmarks—even when we add more rollouts (see the table in our first rebuttal response). The reasons behind this are probably worth further exploration with more experiments; perhaps it’s just not as suitable for the current benchmarks, since most of their inputs are already quite clean.
>
> Instead, our NoisyRollout method really focuses on increasing **rollout diversity** (see Sec. 3.2 in our original manuscript). The goal is to help the model discover more solution patterns and actually learn from them during RL. In that sense, it’s like an RL-specific data augmentation strategy, and this is where our main technical contribution lies.
>
> One last thing we want to clarify: sample-level and rollout-level augmentation are, to some extent, orthogonal. It’s definitely possible to combine both, but making that work well isn’t trivial. We see it as an interesting direction for future work.
>
> We would appreciate it if you could let us know of any further questions or concerns. Thank you again for your review!

---

> ### Comment · Reviewer_5RUf · 2025-08-07
>
> Thanks for confirming that my understanding is correct. My initial concerns were addressed and I have raised the score to 4.
>
> I noticed that the authors conducted "additional experiments evaluating robustness under noisy test conditions on MathVerse" in the rebuttal, which is more aligned with objective 2, if I understand correctly. While it's certainly helpful to show the robustness of the method, it would be helpful to clarify that this is not the main objective of the paper.

---

> ### Author Response · Authors · 2025-08-07
> **Thank you for your support and raising the score**
>
> Thank you very much for your constructive review and encouraging words. We truly appreciate it! You are correct that the experiment on noisy test conditions in MathVerse aligns with objective 2 and was conducted in response to requests from other reviewers. While these results help provide a deeper understanding of our method, and we will consider including them in the final version, as you pointed out, demonstrating improved performance under noisy settings is not the main objective of our work. We will make sure to clarify this point in our final revision and will further polish the paper to incorporate the valuable feedback and insights gained from the rebuttal discussions. Thank you again for your thoughtful review and support!

---

### Official Review · Reviewer_HhCS · 2025-06-30

**Clarity:** 4
**Significance:** 3
**Originality:** 4
**Rating:** 5
**Confidence:** 4

**Summary:**

This paper proposes NoisyRollout, a data augmentation strategy for RL on VLMs, aimed at enhancing visual reasoning robustness. The key idea is to augment rollout diversity by mixing inputs from both clean and moderately distorted images during training. The approach is only a data augmentation with GRPO, requiring minimal extra computation. Extensive experiments across five out-of-domain visual reasoning and perception demonstrate significant performance gains over various RL baselines. The approach also works well across model sizes and data scales. Overall, this is a solid paper. While I do have some detailed design questions hoping authors to justify, the overall contribution is novel and sound.

**Questions:**

See weaknesses.

One more question:
1. The claim that NoisyRollout “scales test-time compute” is potentially confusing—it is not test-time computation per se that is scaled, but policy exploration during training.

**Ethical Concerns:**

["NO or VERY MINOR ethics concerns only"]

**Final Justification:**

I have read all reviews and authors' responses. The authors' response addressed my weaknesses 1-2 very well, and the authors mentioned they will include examples in the final version to address my weakness 3. As such, I increase my rating to 5: accept.

**Limitations:**

Yes

**Quality:**

3

**Strengths And Weaknesses:**

Strengths
1. The paper identifies an underexplored yet important aspect of RL in VLMs—exploration under noisy visual inputs (which itself is a very common computer vision training data augmentation technique).

2. NoisyRollout integrates well with existing GRPO with minimal extra computation just to add noises to the visual inputs. The proposed algorithm is easy to understand and implement.

3. Despite the simplicity, NoisyRollout consistently outperforms other vanilla RL methods. Ablations on noise strength, rollout temperature, annealing, and model size are thorough. The performance gain is general across datasets, architectures, and varying training setups.

Weaknesses
1. It is unclear to me why the authors choose only utilizing the rewards for noisy inputs in calculating the mean in GRPO objective. I would have thought having the “contrastive signal” between clean and noisy rollouts (by using both normal and noisy trajectories) could offer stronger supervision. I hope the authors could provide justification and/or results to support the chosen design option is better.

2. Another design question: why is only Gaussian noise chosen with NoisyRollout? Technically I think many types of noise could work with NoisyRollout and this design option seems not justified and explored.

3. Continuing point 2, while the method clearly improves performance, it would be helpful to include or reference failure cases where the current NoisyRollout hurts where Gaussian noise is not meaningful.

---

> ### Author Rebuttal · Authors · 2025-07-31
>
> Thank you for your supportive review and suggestions. Below we respond to the comments in **Weaknesses (W)** and **Questions (Q)**.
> > ***W1: Why only utilize the rewards for noisy inputs when calculating the mean in the GRPO objective, since having a "contrastive signal" between clean and noisy rollouts could offer stronger supervision?***
> >
>
> You raise a good point about contrastive signals that we agree with: when clean and distorted inputs yield divergent outcomes for the same query, these reward differences serve as **implicit contrastive signals** that refine the model's perceptual behaviors during reasoning (we have provided this discussion in our main text, lines 123-125), which is also evidenced in our perception quality experiments (Figure 3, fourth column).
>
> However, there appears to be a misunderstanding about our implementation regarding the roles of noisy rollouts. Our actual implementation:
>
> - We combine rewards from both clean and noisy rollouts: $r = \lbrace r_i\rbrace_{i=1}^{n_1+n_2}$ includes all rewards from both trajectory types ($n_1$ for clean rollouts, $n_2$ for noisy rollouts)
> - The baseline calculation uses this combined set: $\text{mean}(r)$
> - Advantage computation: $\hat{A}_i = (r_i - \text{mean}(r))/\text{std}(r)$ uses the group baseline
> - Both clean and noisy rollouts contribute to policy updates (**conditioned on clean images**)
>
> > ***W2: why is only Gaussian noise chosen with NoisyRollout?***
> >
>
> Thank you for this important question. First, in Appendix A, we indeed discussed different image augmentation strategies. We previously noted that other augmentation strategies (e.g., rotation, cropping) can easily lead to critical information loss, thereby making the rollouts meaningless or even harmful for policy optimization.
>
> However, after carefully reviewing the image augmentation relevant code in our codebase, we found that our rotation implementation used suboptimal parameters (`expand=False`) that caused information loss. After adopting more appropriate parameters (`image.rotate(max_angle, resample=Image.BILINEAR, expand=True)`), rotation now demonstrates meaningful improvements:
>
> | Method | Image Augmentation | MathVerse | MathVision | MathVista | WeMath | HallusionBench | Average |
> | --- | --- | --- | --- | --- | --- | --- | --- |
> | GRPO | None | 50.8 | 27.3 | 70.5 | 67.4 | 69.8 | 57.2 |
> | GRPO | Gaussian noise | **53.2** | **28.5** | **72.6** | **69.6** | **72.1** | **59.2** |
> | GRPO | Rotation | 52.5 | 28.1 | 71.9 | 68.1 | 70.2 | 58.2 |
>
> This indicates that our NoisyRollout framework can work with various augmentation strategies which would not cause visual information loss. While Gaussian noise remains most effective, likely because it preserves all visual content while adding controlled perturbations, we argue other augmentations can also provide benefits when properly implemented.
>
> > ***W3: Include or reference failure cases where the current NoisyRollout hurts and where Gaussian noise is not meaningful.***
> >
> Thank you for your kind suggestions. We would like to address this concern from two aspects:
>
> **Regarding failure cases:** Due to rebuttal policy limitations, unfortunately we cannot include visual examples here, but will add qualitative analysis of failure cases to our final submission as we believe it would indeed be helpful.
>
> **NoisyRollout degrades to vanilla GRPO performance (rather than hurting it) when visual dependency is minimal**—that is, when reasoning relies primarily on text rather than visual understanding. In such cases, visual augmentation provides no exploration benefit since clean and noisy rollouts yield similar outcomes, causing performance to converge to the baseline (vanilla GRPO). We will include this observation in our Limitations section, as it indeed represents a scenario where NoisyRollout might offer no additional benefit.
>
> > ***Q1: The claim that NoisyRollout “scales test-time compute” is potentially confusing; test-time computation per se is not scaled.***
> >
>
> Thank you for raising this clarification question. In our context, "scaling test-time compute" actually refers to performing reasoning with chain-of-thought (CoT) at test-time. We would like to clarify that our claim is that **NoisyRollout enables more effective CoT reasoning during inference through our noisy RL training instead of using more tokens at test time**. Specifically, NoisyRollout enhances the model's intrinsic reasoning capabilities, allowing it to achieve superior visual reasoning performance when performing CoT at test-time through the improved problem-solving abilities developed via our noisy RL training.

---

> > ### Comment · Reviewer_HhCS · 2025-08-05
> >
> > I have read all reviews and authors' responses. The authors' response addressed my weaknesses 1-2 very well, and the authors mentioned they will include examples in the final version to address my weakness 3. As such, I increase my rating to 5: accept.

---

> > > ### Author Response · Authors · 2025-08-06
> > > **Thank you for your support**
> > >
> > > Thank you for your prompt feedback and encouraging words. We really appreciate it! In our final revision, we will polish the paper further to incorporate the valuable insights gained from the rebuttal discussions. Thank you again!

---

### Official Review · Reviewer_mrqL · 2025-07-02

**Clarity:** 3
**Significance:** 3
**Originality:** 2
**Rating:** 4
**Confidence:** 3

**Summary:**

This paper studies fine-tuning VLMs via reinforcement learning with visual domain shifts. It introduces NoisyRollout, which interleaves on-policy rollouts under additive Gaussian noise with clean rollouts in GRPO update, while annealing the noise strength over training. By always computing policy updates on the clean input, NoisyRollout drives exploration without corrupting the learned input distribution. Experiments shows empirical performance gain across 5 out-of-domain reasoning and perception benchmarks.

**Questions:**

1. On Proportion of noisy rollouts, I wonder if the authors explored adaptive ratios that shift over time or with model confidence?
2. Gradient Contribution Analysis: the paper only report norm magnitudes, but not whether noisy gradients actually improve or sometimes misdirect the update. dominance in magnitude doesn’t definitively prove that noisy gradients *cause* better exploration, it could also be a byproduct of higher reward variance under noise. More in-depth discussions on this will be appreciated.

**Ethical Concerns:**

["NO or VERY MINOR ethics concerns only"]

**Final Justification:**

Post-rebuttal final justification: Thank you for your in-depth rebuttal. Your responses have addressed most my questions and concerns. However, I still think that Gaussian noise has limited novelty and is not broadly applicable to other more complicated domains. Therefore, I will maintain my score as 4: borderline accept.

**Limitations:**

Yes.

**Quality:**

2

**Strengths And Weaknesses:**

**Strengths**:

The paper proposes a simple yet broadly applicable idea of interleaving noisy and clean rollouts without changing the core RL formulation, making it easy to integrate into existing VLM policy fine-tuning pipelines. The experiments and ablations are very thorough and show strong empirical performance gains.


**Weaknesses**

1. Limited novelty: Injecting synthetic distortions to improve robustness is not entirely a new idea, e.g., “domain randomization” in computer vision and robotics. the paper lacks a strong argument why its specific rollout-mixing strategy is meaningfully better  than existing perturbation-based exploration methods.
2. The synthetic noise might not reflect real world situations: The proposed method uses only additive Gaussian noise for all noisy rollouts. This is a very narrow and idealistic corruption model, real-world domain shifts (e.g. occlusion, blur, viewpoint changes, etc.) aren’t covered. If the policy never sees those distortions in rollout, there’s no guarantee it will generalize to them.
3. No sequential task: All experiments are on single-step vision-reasoning. It’s unclear whether hybrid rollouts would benefit longer horizon policies where perception errors compound. A good example could be robotic navigation or manipulation under changing visual conditions.
4. No results on test-time noise: Because updates are always conditioned on the clean image, the policy never actually learns to interpret noisy inputs, so it may still fail at inference when images are added with noise. The only robustness induced is by selecting actions that succeed under Gaussian noise during rollout (more exploration), but we don’t see direct evaluation of the final policy on noisy test images.

---

> ### Author Rebuttal · Authors · 2025-07-31
>
> Thank you for your supportive review and suggestions. Below we respond to the comments in **Weaknesses (W)** and **Questions (Q)**.
> > ***W1: Injecting synthetic distortions to improve robustness is not entirely a new idea; the paper lacks a strong argument why its specific rollout-mixing strategy is meaningfully better than existing perturbation-based exploration methods.***
>
> Thank you for highlighting the importance of clarifying how our rollout-mixing strategy differs from existing perturbation-based methods. We acknowledge that injecting synthetic distortions is an established approach, however, we wish to clarify the specific novelty and unique considerations of our NoisyRollout method:
>
> 1. **Rollout-level vs. sample-level augmentation.** Our method introduces a distinct augmentation strategy tailored specifically for the RLVR and VLM setting:
>     - **Traditional sample-level augmentation** (DrQ [1], RAD [2], etc.): Rollouts are collected from both original and augmented images, with each trajectory optimized according to its actual source image during policy updates.
>     - **Our rollout-level augmentation** (our NoisyRollout): We collect trajectories from both original and augmented images, but during policy optimization, we treat all rollouts as if they originated from clean images, regardless of their actual source, as indicated in Eq. (2).
> 2. **Specific considerations in choosing rollout-level augmentation.** Our rollout-level strategy addresses key characteristics of RLVR with VLMs:
>     - We use noisy rollouts to **increase rollout diversity** and **enhance exploration**, targeting visual perception (a well-known VLM weakness) to provide more diverse and meaningful supervision signals during RL training.
>     - While we focus on rollout-level augmentation for these advantages, sample-level augmentation remains valuable for future exploration, potentially offering complementary strengths.
> 3. **Empirical support.** In the Appendix (lines 616-628) of our original manuscript, we have previously noted that applying augmentation at the sample level is less effective than our proposed rollout-level augmentation, and as part of our rebuttal, we provide an additional quantitative experiment to substantiate this claim:
>     - **Setup**: Trained on the Geometry3K dataset with Qwen2.5-7B-VL-Instruct; `rollout=6+6` denotes 6 rollouts from clean images and 6 rollouts from noisy images
>     - **Observation**: Rollout-level augmentation consistently outperforms both baseline and sample-level augmentation
>     - **Why it works**: Sample-level augmentation restricts exploration since augmented rollouts stay tied to their original image-question pairs ("on-policy"). Rollout-level augmentation enables diverse exploration through visual-oriented inductive bias, helping models benefit from more effective supervision signals brought by noisy rollouts. A recent study in the LLM domain [1] shares a similar observation regarding the effectiveness of "off-policy" rollouts.
>         | Method | Augmentation Type | MathVerse | MathVision | MathVista | WeMath | HallusionBench | Average |
>         | --- | --- | --- | --- | --- | --- | --- | --- |
>         | GRPO | none (rollout=12) | 50.8 | 27.3 | 70.5 | 67.4 | 69.8 | 57.2 |
>         | NoisyRollout | sample-level (rollout=6+6) | 49.5 | 28.4 | 69.3 | 67.4 | 68.6 | 56.6 |
>         | NoisyRollout | sample-level (rollout=12+12) | 50.7 | 26.5 | 70.1 | 67.9 | 70.0 | 57.0 |
>         | NoisyRollout | rollout-level (rollout=6+6) | **53.2** | **28.5** | **72.6** | **69.6** | **72.1** | **59.2** |
>
>         ***Note**: Sample-level augmentation underperforms vanilla GRPO in our experiments. This is partly because we maintained batch consistency with our rollout-level approach by including both original and augmented versions of each image into one batch, rather than treating them like independent samples.*
>
> Reference:
>
> [1]. Yarats, Denis, et al. "Image augmentation is all you need: Regularizing deep reinforcement learning from pixels." (ICLR 2021)
>
> [2]. Laskin, Misha, et al. "Reinforcement learning with augmented data." (NeurIPS 2020)
>
> ---
> > ***W2: Only additive Gaussian noise for all noisy rollouts; real-world domain shifts (e.g. occlusion) aren’t covered.***
>
> Thank you for raising this important limitation regarding the generalizability of our augmentation approach. We address this concern through domain-specific justification and empirical validation:
>
> 1. **Gaussian noise is well-suited for our domain.** Mathematical diagrams and equations typically maintain consistent viewpoints and clear presentation. Gaussian noise naturally models sensor noise and compression artifacts that are more common when digitizing mathematical content, unlike occlusion or viewpoint changes which occur less frequently.
> 2. **Empirical validation with additional augmentation types.** Furthermore, we conducted experiments with rotation augmentation as a representative geometric transformation. Our initial rotation experiments showed limited improvement due to implementation issues—specifically using `expand=False`, which cropped rotated images and removed critical visual information. After correcting this with proper parameters (`image.rotate(max_angle, resample=Image.BILINEAR, expand=True)`), rotation augmentation now demonstrates meaningful gains:
>     | Method | Augmentation | MathVerse | MathVision | MathVista | WeMath | HallusionBench | Average |
>     | --- | --- | --- | --- | --- | --- | --- | --- |
>     | GRPO | None | 50.8 | 27.3 | 70.5 | 67.4 | 69.8 | 57.2 |
>     | NoisyRollout | Gaussian noise | **53.2** | **28.5** | **72.6** | **69.6** | **72.1** | **59.2** |
>     | NoisyRollout | Rotation | 52.5 | 28.1 | 71.9 | 68.1 | 70.2 | 58.2 |
> 3. **Combining augmentations show mixed results and require further investigation.** We also explored combining two augmentation types by randomly selecting between Gaussian noise and rotation for each training sample. However, the results reveal that effective augmentation combination is non-trivial:
>
>
>     | Augmentation | Average |
>     | --- | --- |
>     | Gaussian noise | **59.2** |
>     | Rotation | 58.2 |
>     | Gaussian noise  + Rotation | 58.5 |
>
>     The combination approach outperforms rotation alone but falls short of pure Gaussian noise augmentation. This suggests that simply combining augmentation techniques does not guarantee additive benefits. Understanding how to optimally combine multiple augmentation strategies remains an important direction for future work.
>
>
> ---
>
> > ***W3: No sequential task; Unclear whether hybrid rollouts would benefit longer horizon policies where perception errors compound.***
>
> Thank you for this important observation. We address this concern from two perspectives:
>
> 1. **Our preliminary evidence suggests combining augmentation strategies is non-trivial.** We tested mixed augmentation (Gaussian noise + rotation) and found it underperformed pure Gaussian noise (58.5 vs 59.2 average), indicating that effective combination requires more sophisticated approaches than random selection (see Weakness 2, point 3).
> 2. **Sequential tasks are beyond our current scope.** They would require addressing catastrophic forgetting, entropy collapse, and continual learning complexities—substantial methodological challenges that need dedicated future investigation.
>
> ---
>
> > ***W4: No results on test-time noise; it may still fail at inference when images are added with noise.***
>
> Thank you for this important concern. We conducted additional experiments evaluating robustness under noisy test conditions on MathVerse:
>
> | Method | Clean | Noise Step (300) | Noise Step (500) |
> | --- | --- | --- | --- |
> | GRPO | 50.8 | 50.2 | 44.7 |
> | ThinkLite | 50.2 | 49.6 | 44.3 |
> | **NoisyRollout** | **53.2** | **52.3** | **46.2** |
>
> NoisyRollout consistently outperforms baselines across all noise levels, demonstrating that our training-time augmentation successfully improves test-time robustness to noisy inputs.
>
> ---
>
> > ***Q1: Adaptive ratios that shift over time or with model confidence.***
>
> Thank you for this thoughtful question. That's an good idea that aligns well with our current noise annealing strategy in terms of the training dynamics.
>
> We didn't explore adaptive ratios empirically, but we did consider this approach during early development. Here's our reasoning for the current design choice:
>
> 1. **Fundamental limitation of proportion-based approaches.** Even when adaptively decreasing noisy rollout proportions during training, each individual noisy rollout still introduces the same degree of distributional mismatch. The core problem—distribution shift at the `<input, rollout>` level—remains unresolved regardless of how many noisy rollouts we include.
> 2. **Noise annealing addresses the root cause.** Our approach directly reduces augmentation intensity over time, allowing us to maintain exploration benefits early in training while progressively aligning with the clean image distribution encountered at inference. This manages the exploration-exploitation trade-off more systematically.
>
> ---
>
> > ***Q2: Dominance in magnitude doesn’t definitively prove that noisy gradients cause better exploration, it could also be a byproduct of higher reward variance under noise.***
>
> Thank you for this important methodological critique. You're right that our gradient norm analysis cannot definitively distinguish between beneficial exploration and misdirection from reward variance. While higher magnitudes correlate with improved performance across benchmarks, this doesn't establish causality. Our current analysis provides preliminary insights into the relationship between noisy rollouts and gradient behavior.
>
> Nevertheless, we empirically believe that noisy gradients lead to better exploration, supported by our ablation study on rollout diversity in Section 3.2. More sophisticated rollout attribution analysis would provide deeper insights, which we leave as important future work.

---

> ### Comment · Reviewer_mrqL · 2025-08-05
>
> Dear authors,
>
> Thank you for your in-depth rebuttal. Your responses have addressed most my questions and concerns. However, I still think that Gaussian noise has limited novelty and is not broadly applicable to other more complicated domains. Therefore, I will maintain my score.
>
> Thanks,
>
> Reviewer mrqL

---

> > ### Author Response · Authors · 2025-08-06
> > **Thank you for your feedback**
> >
> > Dear Reviewer mrqL,
> >
> > Thank you so much for your valuable feedback – we really appreciate it! While we've validated the effectiveness of Gaussian noise and rotation in our proposed method, we focused specifically on visual reasoning (especially math problems), so other augmentation strategies like occlusion, blur, and viewpoint changes (which are more common in robotics) weren't covered in this work. It's definitely an interesting direction that's worth exploring further, but it was beyond the scope of our current work.
> >
> > Hopefully, our findings and empirical results can serve as a foundation for future research exploring these additional augmentation strategies. Thanks again for the constructive suggestions!
> >
> > Best,
> >
> > Authors

---

### Official Review · Reviewer_aFBH · 2025-07-03

**Clarity:** 3
**Significance:** 4
**Originality:** 2
**Rating:** 5
**Confidence:** 3

**Summary:**

This paper introduces a simple strategy of augmenting image modality with gaussian noise during the VLM finetuning with RL. The authors show that current VLM methods are sensitive to visually OOD data -- and propose this simple pipeline that is scalable to any RL training.
They conduct experiments on 2 datasets and show significant, non-trivial improvements on OOD perceptual reasoning tasks.

**Questions:**

2. What is the intuition behind starting with a very noisy image and slowly annealing the noise to effectively get a clean image in the later iterations? Intuitively, I'd have assumed that slowly increasing the noise would act as a curriculum.

3. It is indeed very interesting to see that only gaussian noise worked and other data augmentations like cropping or rotation did not. I can see why cropping may hamper the learning process, especially when crucial parts of the image are cropped out. However, it's interesting to see rotation fail as well. Have authors tried another curriculum where they train with gaussian noise, then with rotation / cropping (after letting the model be stable). One potential risk with this could be the ovbious forgetting problem in ML. But I'm curious if the authors had tried this in their experiments.

**Ethical Concerns:**

["NO or VERY MINOR ethics concerns only"]

**Final Justification:**

Thanks to the authors for their comprehensive rebuttal and for performing the additional experiments. I'm glad that the rotation augmentation worked reasonably well after the correct implementation for the augmentation.

Overall, all of my concerns were adequately addressed and I will choose to keep my original rating of **Accept**.

**Limitations:**

Yes.

**Paper Formatting Concerns:**

None.

**Quality:**

3

**Strengths And Weaknesses:**

**Strengths**


- Practical and scalable approach with minimal changes required to the existing RL training framework for VLMs.

- Significant improvement on OOD tasks.

**Weaknesses**

1. The use of data augmentation is a very common choice in RL and has been extensively studied in Visual RL and using them is not really novel (which is fine according to me). Works such as Drq, DrqV2, RAD, CURL [1, 2, 3, 4] should be cited and discusses in the related work section.
---
**References**

[1] Yarats, D., Kostrikov, I., & Fergus, R. (2021). Image Augmentation Is All You Need: Regularizing Deep Reinforcement Learning from Pixels. In 9th International Conference on Learning Representations (ICLR 2021)

[2] Yarats, D., Fergus, R., Lazaric, A., Pinto, L., & Kostrikov, I. (2022). Mastering Visual Continuous Control: Improved Data-Augmented Reinforcement Learning. In International Conference on Learning Representations (ICLR 2022)

[3] Laskin, M., Srinivas, A., & Abbeel, P. (2020). Reinforcement Learning with Augmented Data. In Advances in Neural Information Processing Systems (NeurIPS 2020).

[4] Srinivas, A., Laskin, M., & Abbeel, P. (2020). CURL: Contrastive Unsupervised Representations for Reinforcement Learning. In International Conference on Machine Learning (ICML 2020).

---

> ### Author Rebuttal · Authors · 2025-07-31
>
> Thank you for your supportive review and suggestions. Below we respond to the comments in **Weaknesses (W)** and **Questions (Q)**.
> > ***W1: Data augmentation in RL is common and non-novel; discuss and cite DrQ, DrQv2, RAD, and CURL in related work.***
> >
>
> Thank you for emphasizing the importance of discussing existing RL works that use data augmentation. We recognize that data augmentation is well-established in both visual tasks and reinforcement learning. We will include foundational works like DrQ, DrQ-v2, RAD, and CURL in our revised related work section.
>
> Nevertheless, we would like to further clarify what makes our approach novel:
>
> 1. **Data augmentation designed for RLVR+VLM**. While reinforcement learning with verifiable rewards (RLVR) has shown great success in LLMs (like Tulu3 and DeepSeek-R1), it hadn't been thoroughly tested in VLMs when we started our study. We found that VLMs' weak visual perception prevents them from achieving the same impressive RLVR results as LLMs, which motivated us to use image augmentation to make the model's visual reasoning more robust to the imperfect perception characteristics of VLMs. **To the best of our knowledge, we are the first work to utilize data augmentation in RLVR + VLM.**
>
> 2. **Rollout-level vs. sample-level augmentation**. Despite direct adaptation from traditional data augmentation implementation, our method differs fundamentally:
>     - **Traditional sample-level augmentation** (DrQ, RAD, etc.): Rollouts are collected from both original and augmented images, with each trajectory optimized according to its actual source image during policy updates.
>     - **Our rollout-level augmentation** (NoisyRollout): We collect trajectories from both original and augmented images, but during policy optimization, we treat all rollouts as if they originated from clean images, regardless of their actual source, as indicated in Equation (2).
>
>     In the Appendix (lines 616-628) of our original manuscript, we have previously noted that applying augmentation at the sample level is less effective than our proposed rollout-level augmentation, and as part of our rebuttal, we provide an additional quantitative experiment to substantiate this claim:
>
>     - **Setup**: Trained on Geometry3K dataset (2.1K samples) with Qwen2.5-7B-VL-Instruct; `rollout=6+6` denotes 6 rollouts from clean images and 6 rollouts from noisy images
>     - **Observation**: Rollout-level augmentation consistently outperforms both baseline and sample-level augmentation across all benchmarks
>     - **Why it works**: Sample-level augmentation restricts exploration since augmented rollouts stay tied to their original image-question pairs ("on-policy"). Rollout-level augmentation enables diverse exploration through visual-oriented inductive bias, helping models benefit from more effective supervision signals brought by noisy rollouts. A recent study in the LLM domain [1] shares a similar observation regarding the effectiveness of "off-policy" rollouts.
>
>     | Method | Augmentation Type | MathVerse | MathVision | MathVista | WeMath | HallusionBench | Average |
>     | --- | --- | --- | --- | --- | --- | --- | --- |
>     | GRPO | None (rollout=12) | 50.8 | 27.3 | 70.5 | 67.4 | 69.8 | 57.2 |
>     | NoisyRollout | Sample-level (rollout=6+6) | 49.5 | 28.4 | 69.3 | 67.4 | 68.6 | 56.6 |
>     | NoisyRollout | Sample-level (rollout=12+12) | 50.7 | 26.5 | 70.1 | 67.9 | 70.0 | 57.0 |
>     | NoisyRollout | Rollout-level (rollout=6+6) | **53.2** | **28.5** | **72.6** | **69.6** | **72.1** | **59.2** |
>
>     ***Note**: The above table shows sample-level augmentation underperforms vanilla GRPO. Unlike standard sample-level methods that randomly sample augmented data, we intentionally included both original and augmented versions of the same image within each batch to maintain consistency with our rollout-level approach (which includes all rollouts from a given sample in one batch). This design choice may slightly hurt sample-level performance, though optimizing the sampling strategy is left for future work.*
>
>
> In summary, NoisyRollout introduces a simple, effective rollout-level augmentation strategy that specifically addresses visual reasoning challenges through vision-oriented inductive biases in the emerging RLVR+VLM domain.
>
> [1]. Yan, Jianhao, et al. "Learning to reason under off-policy guidance." (arXiv 2025)
>
> ---
>
> > ***Q2: Why start with maximum noise and gradually reduce it rather than slowly increase noise as curriculum?***
> >
>
> Thank you for this insightful question. Our annealing strategy (starting with high noise and gradually reducing it) is motivated by two key considerations:
>
> 1. **Why annealing is necessary:** Our rollout-level augmentation creates beneficial "off-policy" exploration but can introduce distributional shift. Early in training, when the model has poor visual understanding, it naturally struggles with clean images anyway, so noisy rollouts provide valuable exploration without significant harm.
>
>     However, as training progresses and the model's visual capabilities improve, continued use of highly noisy rollouts becomes counterproductive—the model no longer needs aggressive exploration and the noise primarily introduces unwanted distribution mismatch. Thus, we need to anneal the noise strength to avoid this.
>
> 2. **Why decreasing rather than increasing noise:** We start with high noise when exploration matters most (when the model is weakest), then gradually reduce it to align with the clean image distribution encountered at inference and exploit rewards more effectively at the later training stage.
>
>     Increasing noise would provide minimal exploration benefit early on but cause maximum distributional harm later, leading to training divergence. Since Figure 5 shows that maintaining consistent noise strength causes divergence in later stages, an "increasing noise" approach would likely suffer from the same problem but to a greater extent.
>
>     However, we think slowly increasing noise could be implemented in sample-level augmentation, since there is no distributional shift issue as it's fully "on-policy".
>
>
> **In essence**, we use high noise when exploration matters most, then reduce it as exploitation becomes more important.
>
> ---
>
> > ***Q3: Interestingly only Gaussian noise worked; tried curriculum training with Gaussian noise first, then rotation/cropping?***
> >
>
> We further investigated this phenomenon and here are our responses to the points you proposed on image augmentation:
>
> 1. **Rotation now works with proper implementation.** Our initial rotation experiments used suboptimal parameters (`expand=False`) that cropped rotated images and lost critical visual information. With corrected parameters (`image.rotate(max_angle, resample=Image.BILINEAR, expand=True)`), rotation now shows meaningful improvements:
>
>
>     | Method | Image Augmentation | MathVerse | MathVision | MathVista | WeMath | HallusionBench | Average |
>     | --- | --- | --- | --- | --- | --- | --- | --- |
>     | GRPO | None | 50.8 | 27.3 | 70.5 | 67.4 | 69.8 | 57.2 |
>     | NoisyRollout | Gaussian noise | **53.2** | **28.5** | **72.6** | **69.6** | **72.1** | **59.2** |
>     | NoisyRollout | Rotation | 52.5 | 28.1 | 71.9 | 68.1 | 70.2 | 58.2 |
>
>     Both augmentation types now outperform vanilla GRPO, confirming that NoisyRollout benefits from various augmentations when properly tuned.
>
> 2. **Sequential curriculum training faces entropy collapse challenges.** Your proposed approach of starting with Gaussian noise then transitioning to rotation/cropping is compelling, and we had similar considerations early in our research. However, we identified a fundamental concern: potential entropy collapse after the initial Gaussian noise stage could make the model's policy too deterministic, limiting subsequent rotation-based training effectiveness. This would require non-trivial technical solutions, preventing us from pursuing this direction.
>
> 3. **Random augmentation selection yields modest gains over rotation but falls short of Gaussian noise.** Instead, we explored randomly selecting between Gaussian noise and rotation for each sample during training (that is, the training compute still keeps the same among the three experimental settings):
>
>
>     | Image Augmentation | Average |
>     | --- | --- |
>     | Gaussian noise | **59.2** |
>     | Rotation | 58.2 |
>     | Gaussian noise  + Rotation | 58.5 |
>
>     While the combination performs better than rotation alone, it doesn't exceed single Gaussian noise augmentation, suggesting that combining different augmentations effectively is non-trivial and needs further investigation.
>
> Due to computational constraints, we reserve the systematic exploration of curriculum-based augmentation strategies for future work.

---

> > ### Comment · Reviewer_aFBH · 2025-08-09
> > **Thanks for the rebuttal**
> >
> > [Reposting this since I'd already updated my Final Justification but realized that it won't be open to authors until the decisions come out]
> >
> > Thanks to the authors for their comprehensive rebuttal and for performing the additional experiments. I'm glad that the rotation augmentation worked reasonably well after the correct implementation for the augmentation.
> >
> > Overall, all of my concerns were adequately addressed and I will choose to keep my original rating of Accept.

---

> > > ### Author Response · Authors · 2025-08-09
> > > **Thank you for your response**
> > >
> > > Thank you for your encouraging feedback and for letting us know your decision on the final rating! We’re glad the clarifications and additional experiments addressed your concerns (especially the corrected rotation augmentation). In the final revision, we will further refine the paper and incorporate insights from the rebuttal discussions. We truly appreciate your constructive review and support.

---

### Note · Authors · 2025-08-13

Dear Area Chairs and Reviewers,

We sincerely thank you for your time, thoughtful comments, and constructive engagement throughout the review process. We greatly appreciate the reviewers' recognition of our work's simplicity, effectiveness, and empirical strength.

In our response, we have addressed all reviewers' questions, clarified potential misunderstandings, and conducted additional experiments to strengthen our findings:

**1. Rollout-Level Augmentation**
   * **Key distinction from traditional approaches**: Our method collects rollouts from both clean and noisy images but crucially performs policy optimization on clean inputs only, treating all rollouts as if from clean images. This enables "off-policy" exploration with diverse supervision signals.
   * **Provided quantitative evidence** showing rollout-level augmentation consistently outperforms sample-level augmentation across all benchmarks (59.2 vs 56.6 average score).
   * **Demonstrated we are the first** to apply data augmentation in the RLVR+VLM setting (with a specialized rollout augmentation design), addressing VLMs' weak visual perception that prevents them from achieving impressive RLVR results like LLMs.

**2. Beyond Gaussian Noise**
   * **Fixed rotation implementation issues** (corrected expand=True parameter) and showed rotation augmentation now achieves meaningful improvements.
   * **Explored combined augmentation strategies** (Gaussian noise + rotation), revealing that effective combination is non-trivial and requires further investigation.
   * **Validated test-time robustness**: NoisyRollout consistently outperforms baselines under noisy test conditions.

**3. Explanations**
   * **Noise annealing rationale**: High noise early for exploration, gradual reduction for exploitation.
   * **Learning mechanism**: Noisy rollouts provide challenging conditions and diverse solutions, with both successes and failures contributing to robust reasoning.

**4. Clarifications**
   * **Implementation details**: Both clean and noisy rollouts contribute to reward baselines and policy updates.
   * **Perception vs. reasoning**: While challenging to separate strictly in VLM contexts, we acknowledge this merits further exploration in Explainable AI.

Thank you again for your valuable feedback. We will incorporate these insights and additional suggestions (e.g., citing relevant papers in the Visual RL domain and providing better qualitative analysis) into our final revision and release code/models.

---

### Decision · Program_Chairs · 2025-09-17

**Decision:**

Accept (poster)

**Comment:**

This paper addresses visual reasoning in vision-language models (VLMs), a currently important area where recent models struggle with imperfect visual perception that affects reasoning processes. The proposed NoisyRollout method introduces a rollout-level data augmentation strategy that mixes trajectories from clean and noisy images during reinforcement learning training, representing a novel adaptation for the emerging RLVR + VLM setting.

Reviewers consistently recognized the method's simplicity and broad applicability with minimal computational overhead, strong empirical performance gains across multiple benchmarks, and thorough experimental evaluation including comprehensive ablations across model sizes and data scales.

The primary concerns centered on limited novelty since data augmentation in RL is well-established, questions about restricting to Gaussian noise rather than diverse augmentation strategies, and lack of evaluation on sequential tasks or complex real-world domain shifts. Some reviewers also sought clarification on specific design choices like rollout-level versus sample-level augmentation.

The authors provided a comprehensive rebuttal that effectively addressed these concerns by demonstrating quantitatively that rollout-level augmentation outperforms sample-level approaches, showing generalizability beyond Gaussian noise, providing test-time robustness experiments, and clarifying technical distinctions from traditional data augmentation.

While novelty is incremental, the paper makes a solid contribution to the currently high-profile field of visual reasoning in VLMs. Reasons for acceptance include: practical effectiveness and ease of adoption, consistent empirical improvements across diverse benchmarks, thorough experimental validation, and timely contribution to addressing known VLM visual perception weaknesses that limit reasoning performance.